# p53 suppresses MHC class II presentation by intestinal epithelium to protect against radiation-induced gastrointestinal syndrome

Jianming Wang[1], Chun-Yuan Chang [1], Xue Yang[1], Fan Zhou[1], Juan Liu[1], Jill Bargonetti [2], Lanjing Zhang [1,3,4], Ping Xie [1,5], Zhaohui Feng [1] ✉ & Wenwei Hu [1] ✉

Radiation-induced gastrointestinal syndrome is a major complication and limiting factor for radiotherapy. Tumor suppressor p53 has a protective role in radiation-induced gastrointestinal toxicity. However, its underlying mechanism remains unclear. Here we report that regulating the IL12-p40/MHC class II signaling pathway is a critical mechanism by which p53 protects against radiation-induced gastrointestinal syndrome. p53 inhibits the expression of inflammatory cytokine IL12-p40, which in turn suppresses the expression of MHC class II on intestinal epithelial cells to suppress T cell activation and inflammation post-irradiation that causes intestinal stem cell damage. Anti-IL12-p40 neutralizing antibody inhibits inflammation and rescues the defects in intestinal epithelial regeneration post-irradiation in p53-deficient mice and prolongs mouse survival. These results uncover that the IL12-p40/MHC class II signaling mediates the essential role of p53 in ensuring intestinal stem cell function and proper immune reaction in response to radiation to protect mucosal epithelium, and suggest a potential therapeutic strategy to protect against radiation-induced gastrointestinal syndrome.

Radiotherapy is one of the most commonly used cancer therapies, along with surgery and chemotherapy[1]. Radiotherapy targets rapidly proliferating tumor cells and causes DNA damage in these cells, which limits their proliferation and induces cell death[2]. However, the therapeutic benefit of radiotherapy is often limited primarily due to its high toxicity in normal tissues[3]. Rapidly proliferating cells in normal tissues, especially cells in the hematopoietic system and Lgr5+ active intestinal stem cells (ISCs) in the gastrointestinal (GI) tract, are often damaged by radiation[3]. The intestinal epithelium is a dynamic tissue continuously replenished with new cells differentiated from ISCs at the bottom of the intestinal crypts under physiological conditions[2]. Upon injury, ISCs undergo rapid self-renewal and proliferation to maintain the regeneration of the intestinal epithelium[4]. It has been shown that high-dose total body irradiation (TBI) causes the damage of ISCs, leading to the loss of ISCs and impaired function of ISCs, and thus causes severe injury to the GI tract. The severe injury of the GI tract then results in diarrhea, bacterial infection, and malabsorption, which lead to lethality within 10 days post-TBI in mice, a phenomenon known as radiation-induced GI syndrome[3]. Unlike hematopoietic injury that can be rescued by bone marrow (BM) transplantation, radiation-induced GI syndrome is a lethal disorder[5]. There are no effective treatments for radiation-induced GI syndrome so far.

In addition to its routine function in food digestion and nutrient absorption, recent studies have suggested intestinal epithelial cells (IECs) as an important type of antigen-presenting cells (APCs) by expressing major histocompatibility complex (MHC) class II (MHC-II)

[1]Rutgers Cancer Institute of New Jersey, Rutgers University, New Brunswick, NJ 08903, USA. [2]Department of Biological Sciences, Hunter College, City University of New York, New York, NY 10065, USA. [3]Department of Biological Sciences, Rutgers University, Newark, NJ 07102, USA. [4]Department of Pathology, Penn Medicine Princeton Medical Center, Plainsboro, NJ 08536, USA. [5]Department of Cell Biology and Neuroscience, Rutgers University, Piscataway, NJ 08854, USA. ✉e-mail: fengzh@cinj.rutgers.edu; wh221@cinj.rutgers.edu

on the cell surface, which in turn activates T cells[6,7]. The antigen-presenting function of IECs is essential in regulating many pathological processes, such as graft-versus-host disease (GVHD), an allogeneic donor T cell-induced inflammatory disease[7], microbiota-triggered inflammation[8], and intestinal tumorigenesis[6]. However, the involvement of MHC-II on IECs in radiation-induced ISC damage and GI syndrome is currently largely unknown.

Tumor suppressor p53 is one of the major regulators mediating the biological response to radiation[9]. Upon irradiation, p53 senses the DNA damage caused by radiation and transcriptionally activates its downstream targets to regulate many important cellular processes, including cell cycle arrest, senescence, and apoptosis, in a highly cell- and tissue-dependent manner[10]. This leads to target-specific outcomes, including growth suppression in tumor cells and toxicity in normal tissues[10]. Intriguingly, in vivo studies have reported dual effects of p53 in different tissues in response to radiation[2,11]. Upon low-dose irradiation, p53 sensitizes the hematopoietic cells to promote the hematopoietic injury, whereas upon high-dose irradiation, p53 functions as a barrier to reduce the GI injury[2,11]. While early studies have suggested a protective role of p53 in radiation-induced GI toxicity, its precise mechanisms are far from clear, which hinders the development of effective strategies to prevent and treat radiation-induced GI toxicity in clinics.

Here, we report a mechanism of p53 in protecting against radiation-induced ISC damage by limiting the antigen-presenting function of IECs to inhibit the inflammation triggered by TBI. p53 reduces the expression of IL12-p40, an inflammatory cytokine, which in turn suppresses the expression of MHC-II on IECs, leading to reduced inflammation and ISC damage post-TBI. The down-regulation of IL12-p40 levels by p53 is through transcriptional induction of a p53 target gene, leukemia inhibitory factor (LIF). In addition to down-regulating IL12-p40 expression, LIF also protects ISCs from T cell-induced damage and promotes the regeneration of ISCs, contributing to the protective effect of p53 on radiation-induced GI syndrome. Administering anti-IL12-p40 neutralizing antibody (aIL12-p40 antibody) or recombinant LIF (rLIF) protein greatly reduces the ISC damage and inflammation in the gut induced by radiation in p53-deficient mice. Results from this study reveal crucial roles and mechanisms of p53 in supporting ISC function and ameliorating inflammation to protect against radiation-induced GI syndrome, highlighting its potential to alleviate GI toxicity when incorporated with radiotherapy.

## Results

### p53 deficiency in mice leads to a more severe GI syndrome with exacerbated ISC damage and inflammatory immune response

Upon exposure to high-dose TBI, the intestinal epithelium of mice undergoes an apoptotic phase within 36 h post-TBI, followed by a rapid regeneration phase mediated by ISCs[2,3]. p53 has been reported to exert a protective effect on radiation-induced GI syndrome[2,11,12]. Consistent with previous reports[11,12], while p53[+/+] mice subjected to 12 Gy TBI had a median lifespan of 9 days due to the GI syndrome, p53[−/−] mice had a significantly reduced lifespan, with a median lifespan of 5 days (Fig. 1a). Transcriptomic analysis by RNA sequencing (RNA-seq) using the small intestine (SI) from mice at 1 day post-TBI indicated that, compared with p53[+/+] mice, p53[−/−] mice had a more severely impaired intestinal epithelial homeostasis and enhanced immune activation in the SI post-TBI (Fig. 1b). p53[+/+] mice exhibited numerous apoptotic cell deaths in the ISC niche at 1 day post 12 Gy TBI as examined by TUNEL staining, and TBI-induced apoptosis was largely diminished at later time points as examined at both 3 days and 5 days post-TBI, when ISCs repopulated (Fig. S1a), which is consistent with a previous report[13]. In contrast, apoptosis was very limited in p53[−/−] mice at 1 day post 12 Gy TBI, reflecting the loss of p53-mediated apoptosis (Fig. S1a). However, significant cell death due to mitotic catastrophe, another form of cell

death occurring during mitosis that can lead to apoptosis[14,15], was observed in the ISC niche in p53[−/−] mice starting from 3 days post-TBI, and was still observed at 5 days post-TBI (Fig. S1a, b), suggesting increased damage and impaired regeneration of ISCs at later time points post-TBI in p53[−/−] mice. Histopathological analysis using the H&E staining showed that at 1 day post-TBI, there was a great reduction of ISC niches reflected by reduced crypt depth and epithelial damage reflected by disorganized villous architecture due to p53-dependent apoptosis in p53[+/+] mice, while no obvious change was observed in p53[−/−] mice (Fig. 1c). At 3 days post-TBI, while numerous enlarged/hyperplastic crypts indicative of regeneration were observed in p53[+/+] mice, much fewer enlarged/hyperplastic crypts were observed in p53[−/−] mice (Fig. 1c). In turn, at 3 days post-TBI, compared with the SI of p53[+/+] mice, the SI of p53[−/−] mice displayed more severe epithelial injury, including villous blunting and disorganized villous architecture (Fig. 1c). The epithelial injury in p53[−/−] mice was further exacerbated at a later time point (5 days post-TBI) (Fig. 1c). We further examined the number of viable ISCs in mice post-TBI using the immunofluorescence (IF) staining of Olfm4, an ISC marker[4]. Consistent with the epithelial damage and the enhanced number of TUNEL[+] apoptotic cells in the SI of p53[+/+] mice at 1 day post-TBI (Fig. 1c and S1a), the number of Olfm4[+] viable ISCs was greatly reduced in p53[+/+] mice at 1 day post-TBI, which gradually repopulated at 3 days and 5 days post-TBI in p53[+/+] mice (Fig. S2). In contrast, while the number of Olfm4[+] viable ISCs was largely conserved in p53[−/−] mice at 1 day post-TBI due to the lack of apoptosis, ISCs were drastically diminished at 3 days and 5 days post-TBI in p53[−/−] mice (Fig. S2), reflecting the regeneration defect of ISCs in p53[−/−] mice. The significantly reduced number of Olfm4[+] viable ISCs in p53[−/−] mice was further validated using the immunohistochemistry (IHC) staining of Olfm4 at 3 days post-TBI (Fig. 1d). Further, compared with p53[+/+] mice, the number of surrounding supportive Paneth cells was significantly reduced in p53[−/−] mice at 3 days post-TBI as examined by the IF staining of the lysozyme, a Paneth cell marker[4] (Fig. S3). These results indicate that p53 deficiency impairs ISC function and the regeneration of the intestinal epithelium post-TBI.

Radiation is often immunosuppressive by triggering the death of radio-sensitive lymphocytes[16,17]. Indeed, p53[+/+] mice exhibited a significantly diminished number of CD45[+] immune cells and CD3[+] T cells in the SI post-TBI as examined by IF staining assays, and this reduction was much less pronounced in p53[−/−] mice (Figs. S2 and S4). Consistently, a reduction of immune cells, including CD45[+] immune cells, CD4[+] T cells, CD8[+] T cells, and macrophages in mesenteric lymph nodes (MLNs), a crucial lymphoid organ regulating the immune response in the SI[18], was observed in p53[+/+] mice post-TBI by flow cytometric assays (Fig. 2a). The gating strategies of flow cytometric assays are shown in Fig. S5. Notably, no obvious reduction of immune cells in MLNs was observed in p53[−/−] mice post-TBI (Fig. 2a). It is worth noting that the number of dendritic cells (DCs), a radio-resistant immune cell type[19,20], in MLNs was not obviously affected by TBI in p53[+/+] or p53[−/−] mice (Fig. 2a).

TBI has been shown to induce MHC-II expression on IECs[7]. Consistent with previous reports[7,20], the expression of MHC-II on IECs in both villi and crypts in the SI was increased in p53[+/+] mice post-TBI as determined by flow cytometric and IF staining assays, respectively (Figs. 2b and S6). Notably, the induction of MHC-II on IECs was much more pronounced in the SI of p53[−/−] mice post-TBI (Figs. 2b and S6). In line with the higher MHC-II levels on IECs in p53[−/−] mice post-TBI, the number of CD25[+]/CD69[+] activated CD4[+] and CD8[+] T cells in the lamina propria (LP), the connective tissue underlying the intestinal epithelium and a major immune compartment of the SI[21], was significantly higher in p53[−/−] mice compared with p53[+/+] mice post-TBI as determined by flow cytometric assays (Fig. 2c). Notably, the expression levels of MHC-II on other APCs, including macrophages and DCs in the LP, were not significantly affected by TBI and p53 (Fig. S7), highlighting the essential role of antigen presentation by IECs in mediating the inflammatory

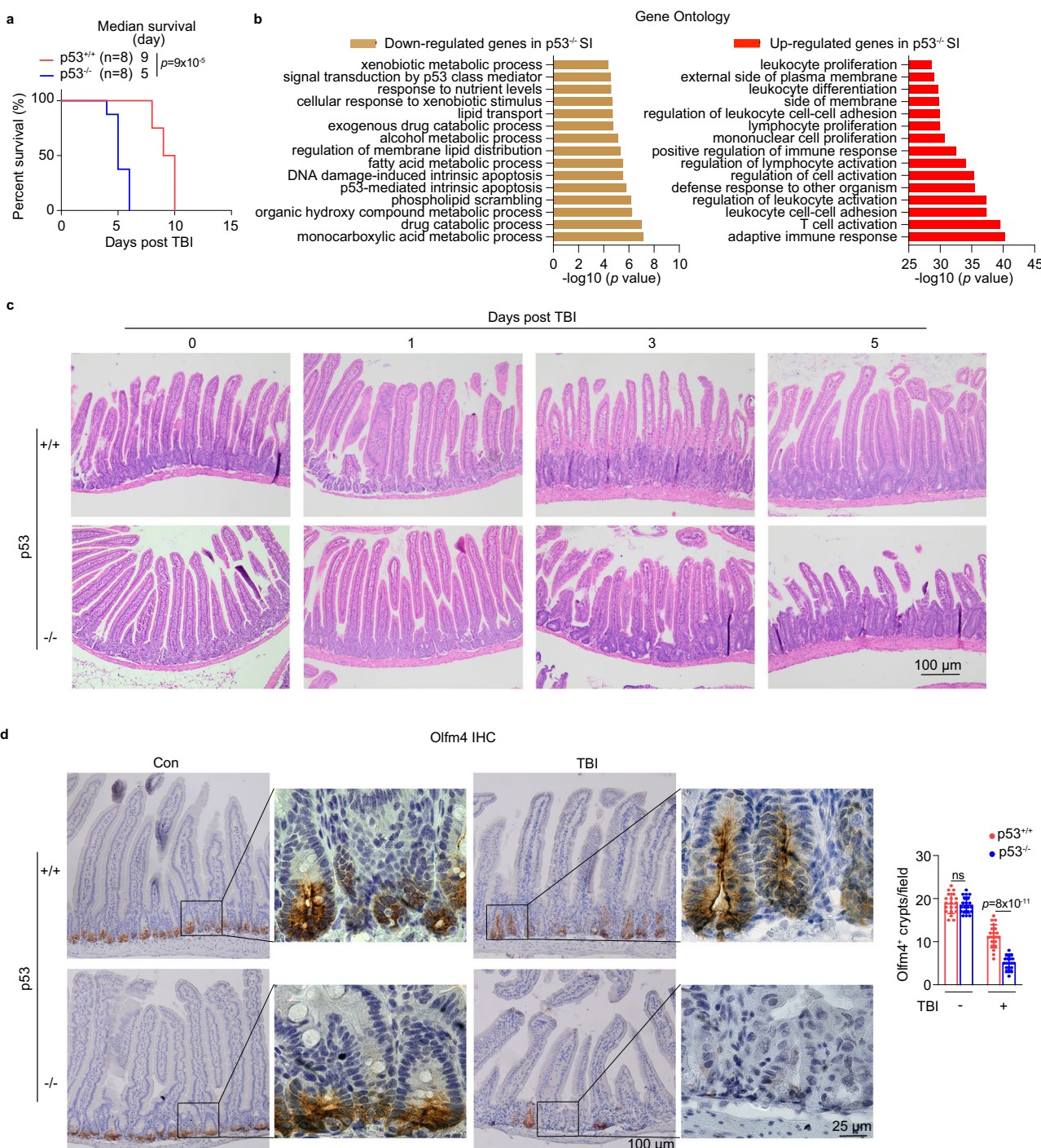

**Fig. 1 | p53 deficiency impairs ISC homeostasis in mice post 12 Gy TBI. a** Kaplan-Meier survival curves of p53$^{+/+}$ and p53$^{-/-}$ mice post-TBI. **b** Gene Ontology (GO) analysis of RNA-seq data. Bar plots show the *p*-value for GO term enrichment of differentially expressed genes between the SI from p53$^{+/+}$ and p53$^{-/-}$ mice at 1 day post-TBI. **c** Representative H&E images from at least 3 independent mice showing the morphology of the SI of mice at different days post-TBI. **d** p53 deficiency impaired ISC homeostasis post-TBI. Representative images from at least 3 independent mice (left) and quantification (right) of IHC staining of Olfm4 in the SI of mice at 3 days post-TBI. *n* = 20 fields from at least 3 mice/group. For (**d**) data are presented as mean ± SD from at least 3 independent experiments. ns: not significant, Kaplan-Meier survival analysis for (**a**) and two-tailed Student's *t*-test for (**d**). Source data are provided as a Source Data file.

condition in p53$^{-/-}$ mice post-TBI. TNFα is an important inflammatory cytokine secreted by activated APCs and T cells to trigger the ISC damage[22,23]. Compared with p53$^{+/+}$ mice, p53$^{-/-}$ mice displayed significantly higher *TNFα* mRNA levels in the SI post-TBI as determined by quantitative real-time PCR (qPCR) assays (Fig. 2d), suggesting an elevated inflammatory immune response in p53$^{-/-}$ mice post-TBI, which is consistent with the transcriptomic profile of the SI from p53$^{+/+}$ and

p53$^{-/-}$ mice post-TBI (Fig. 1b). We further examined the TNFα secretion from T cells and APCs in the LP from p53$^{+/+}$ and p53$^{-/-}$ mice post-TBI using flow cytometric assays. TBI induced significantly higher levels of TNFα secretion from CD4$^+$ and CD8$^+$ T cells, but not macrophages and DCs, in the LP of p53$^{-/-}$ mice compared with p53$^{+/+}$ mice (Fig. S8). Anti-CD3 antibody (aCD3) can effectively deplete CD3$^+$ T cells in mice[24,25]. To investigate whether the increased number of activated T cells and

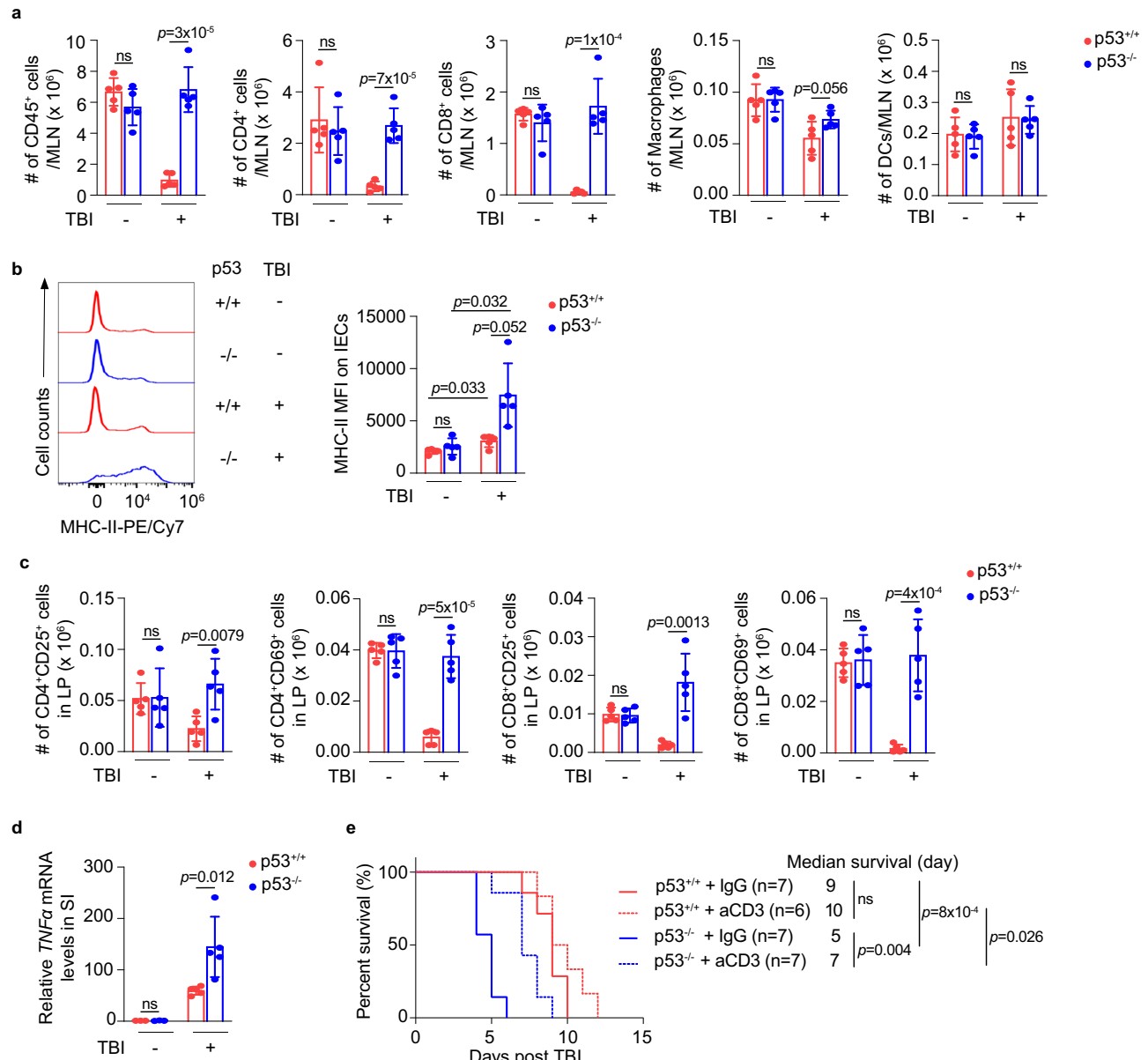

**Fig. 2 | p53 deficiency activates the inflammatory immune response in mice post 12 Gy TBI. a** The number of CD45⁺, CD4⁺, CD8⁺ cells, macrophages and DCs in the MLNs of mice at 1 day post-TBI as determined by flow cytometric assays. Gating strategies are shown in Fig. S5. $n$ = 5 mice/group. **b** Representative histograms from at least 3 independent mice (left) and quantifications of mean fluorescence intensity (MFI) (right) of MHC-II on IECs from mice at 3 days post-TBI as determined by flow cytometric assays. $n$ = 5 mice/group. **c** The number of CD25⁺ or CD69⁺ activated CD4⁺ (left two panels) and CD8⁺ (right two panels) T cells in the LP of mice at 3 days post-TBI as determined by flow cytometric assays. $n$ = 5 mice/group. **d** The

mRNA levels of *TNFα* normalized with the *actin* gene in the SI of mice at 3 days post-TBI as determined by qPCR assays. $n$ = 5 and 3 mice/group for groups with and without TBI treatment, respectively. The *TNFα* mRNA levels in p53⁺/⁺ mice without TBI are designated as 1. **e** Kaplan-Meier survival curves of mice treated with IgG or aCD3 antibodies (200 μg/mouse, once at 6 days before TBI) post 12 Gy TBI. For (**a**–**d**) data are presented as mean ± SD from at least 3 independent experiments. ns: not significant, two-tailed Student's *t*-test for (**a, c, d**) two-tailed Student's *t*-test followed by Bonferroni correction for (**b**) and Kaplan-Meier survival analysis followed by Bonferroni correction for **e**. Source data are provided as a Source Data file.

elevated TNFα production are simply a consequence of reduced T cell apoptosis due to p53 deficiency or actually reflect an enhanced T cell-induced inflammatory immune response that contributes to the more severe TBI-induced GI syndrome in p53⁻/⁻ mice, p53⁺/⁺ and p53⁻/⁻ mice with depletion of T cells using aCD3 treatment were subjected to 12 Gy TBI, and their survival was compared. The depletion of CD3⁺ T cells in the MLN and LP was verified using flow cytometric assays (Fig. S9). Notably, aCD3 treatment significantly prolonged the lifespan in p53⁻/⁻ mice post-TBI, but showed no significant effect on the survival of p53⁺/⁺ mice (Fig. 2e), suggesting that the enhanced T cell-mediated inflammatory immune response contributes to a more severe TBI-induced GI

syndrome observed in p53⁻/⁻ mice. Together, these results suggest that p53 deficiency in mice leads to a much more severe radiation-induced GI syndrome with exacerbated ISC damage and inflammatory immune response.

**p53-deficient ISCs are more vulnerable to T cell-induced damage**
While it is known that p53-mediated damage response sensitizes ISCs towards radiation-induced injury[2,12], the role of p53 in ISCs upon T cell attack is unclear. p53⁻/⁻ mice had significantly increased T cell activation and ISC damage post-TBI (Figs. 1 and 2), but it is unclear whether the increased ISC damage in p53⁻/⁻ mice is due to increased

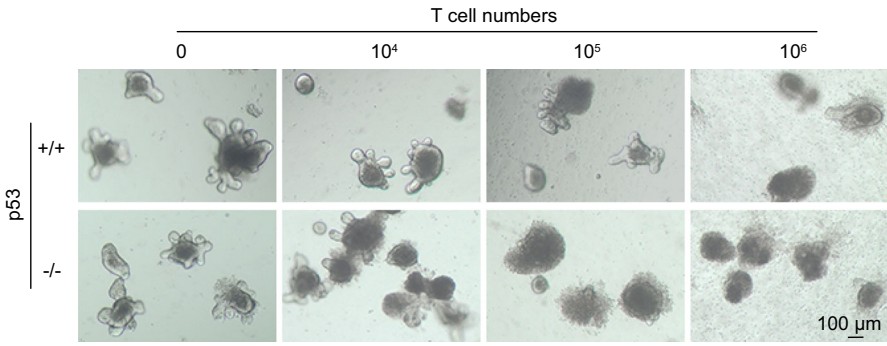

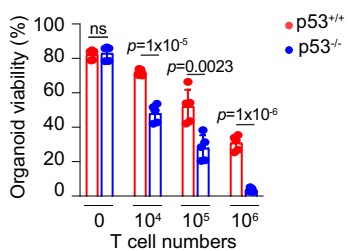

**Fig. 3 | Intestinal organoids derived from p53$^{-/-}$ mice are more susceptible to T cell-induced damage and cell death compared with p53$^{+/+}$ mice.** Left: representative images from at least 3 independent mice showing organoids from p53$^{+/+}$ and p53$^{-/-}$ mice co-cultured with activated T cells; right: quantification of organoid viability measured at 2 days post co-culture using the CyQUANT™ LDH Cytotoxicity Assay Kit. $n = 5$/group. Data are presented as mean ± SD from at least 3 independent experiments. ns: not significant, two-tailed Student's $t$-test. Source data are provided as a Source Data file.

susceptibility of ISCs towards activated T cell-induced damage upon p53 deficiency or solely due to increased T cell activation. To examine whether p53 loss in ISCs leads to increased vulnerability towards T cell-induced damage, we employed a well-established intestinal organoid-T cell co-culture model to assess activated T cell-induced ISC damage in vitro[26]. T cells isolated from p53$^{+/+}$ mice were activated in vitro, followed by co-culturing with intestinal organoids derived from p53$^{+/+}$ and p53$^{-/-}$ mice to induce apoptosis in organoids[27]. T cells reduced the viability of p53$^{+/+}$ and p53$^{-/-}$ organoids in a dose-dependent manner (Fig. 3). Notably, compared with p53$^{+/+}$ organoids, p53$^{-/-}$ organoids were much more susceptible to the challenge of T cells; co-culture with 10$^4$ activated T cells led to ~20% reduction of the viability in p53$^{+/+}$ organoids and over 50% reduction of the viability in p53$^{-/-}$ organoids, and co-culture with 10$^6$ activated T cells led to the death of almost all of the p53$^{-/-}$ organoids, and significantly less death of p53$^{+/+}$ organoids (Fig. 3). These results suggest that p53 deficiency in ISCs increases the ISC damage induced by activated T cells.

**Adoptive transfer of p53$^{-/-}$ BM into p53$^{+/+}$ mice exacerbates the inflammatory immune response and ISC damage post-TBI**

To further characterize the contribution of p53 in immune cell-mediated damage post-TBI, we performed adoptive transfer using the BM derived from p53$^{+/+}$ or p53$^{-/-}$ mice to transplant into lethally irradiated p53$^{+/+}$ recipients (Fig. 4a). This enables the generation of chimeric mice with p53$^{+/+}$ recipient ISCs along with p53$^{+/+}$ or p53$^{-/-}$ donor immune cells. The reconstitution of donor CD45$^+$ immune cells was comparable in CD45.1$^+$ mice receiving BM derived from CD45.2$^+$ p53$^{+/+}$ and p53$^{-/-}$ mice as determined at 28 days after adoptive transfer by flow cytometric assays (Fig. S10a). The recipient mice were subjected to 12 Gy TBI at 28 days after adoptive transfer. A clear reduction of immune cells at 3 days post-TBI was observed in mice receiving p53$^{+/+}$ BM; TBI significantly reduced the number of immune cells, including CD45$^+$ cells, CD4$^+$ T cells, CD8$^+$ T cells and macrophages in MLNs post-TBI (Fig. 4b), which recapitulated the reduced number of immune cells in p53$^{+/+}$ mice post-TBI (Fig. 2a). The reduction in the number of these immune cells in MLNs post-TBI was much less pronounced or not obvious in mice receiving p53$^{-/-}$ BM compared with mice receiving p53$^{+/+}$ BM (Fig. 4b). Mice receiving p53$^{-/-}$ BM also had a significantly higher number of CD25$^+$ activated CD4$^+$ and CD8$^+$ T cells in the LP, and significantly higher levels of *TNFα* expression in the SI at 5 days post-TBI than mice receiving p53$^{+/+}$ BM (Fig. 4c, d), reflecting increased immune activation and inflammation in mice receiving p53$^{-/-}$ BM. In turn, mice receiving p53$^{-/-}$ BM had a much higher number of apoptotic cells in the SI as examined by TUNEL assays, and a much lower number of viable Olfm4$^+$ ISCs and lysozyme$^+$ Paneth cells at 5 days post-TBI compared with mice receiving p53$^{+/+}$ BM (Figs. 4e, f and S10b). These

results suggest that p53 deficiency in immune cells results in their resistance to TBI-induced cell death at an early time point post-TBI to enhance the inflammatory response, leading to more severe ISC damage at later time points post-TBI.

**p53 deficiency induces IL12-p40 expression in DCs to exacerbate GI damage post-TBI**

To investigate the underlying mechanism by which p53 regulates the immune response post-TBI, we compared cytokine levels in the serum between p53$^{+/+}$ and p53$^{-/-}$ mice at 3 days post-TBI using a Proteome Profiler Mouse XL Cytokine Array that contains 111 cytokines involved in various biological processes. While the levels of the majority of cytokines in the serum were comparable between p53$^{+/+}$ and p53$^{-/-}$ mice post-TBI, the levels of several cytokines, including IL12-p40, Fms related receptor tyrosine kinase 3 ligand (Flt3lg) and epidermal growth factor (EGF), showed differences (Fig. 5a). Among them, IL12-p40, a subunit of IL12, a well-known inflammatory cytokine[28], showed the most obvious increase in the serum of p53$^{-/-}$ mice than p53$^{+/+}$ mice post-TBI, suggesting a potential role of IL12-p40 in the more pronounced inflammation observed in p53$^{-/-}$ mice post-TBI (Fig. 5a). The higher serum levels of IL12-p40 in p53$^{-/-}$ mice than p53$^{+/+}$ mice post-TBI were confirmed by ELISA assays (Fig. 5b). The difference in IL12-p40 levels between p53$^{-/-}$ and p53$^{+/+}$ mice was also observed in the SI post-TBI at both mRNA and protein levels determined by qPCR and ELISA assays, respectively (Fig. 5c). While the basal mRNA levels of *IL12b*, which encodes IL12-p40, and the basal protein levels of IL12-p40 were comparable between p53$^{+/+}$ and p53$^{-/-}$ mice in the SI, p53$^{-/-}$ mice displayed much higher *IL12b* and IL12-p40 levels in the SI than p53$^{+/+}$ mice post-TBI (Fig. 5c).

DCs are a major type of immune cells that produce IL12-p40 post-TBI[20,29]. To study whether p53 regulates IL12-p40 expression in DCs, we employed BM-derived DCs (BMDCs), a widely-used model to study DC function in vitro[30]. Upon stimulation by IFNγ and lipopolysaccharides (LPS) to mimic the TBI-induced inflammatory microenvironment, BMDCs derived from both p53$^{+/+}$ and p53$^{-/-}$ mice showed a dramatic elevation of *IL12b* mRNA levels, and notably, the induction of *IL12b* expression was significantly higher in p53$^{-/-}$ BMDCs than p53$^{+/+}$ BMDCs (Fig. 5d), suggesting that p53 deficiency promotes IL12-p40 production in DCs upon inflammatory stimulation. This finding was further validated by employing the IL12-p40-IRES-eYFP (IL12p40-YFP) mouse model, which has a YFP knock-in allele at the IL12-p40 locus to enable the detection of IL12-p40 using YFP[31]. The E3 ubiquitin ligase MDM2 is a key negative regulator of p53, which binds to p53 and promotes the ubiquitination and degradation of p53[32]. RG7112, a small-molecule antagonist of MDM2 that inhibits the MDM2-p53 binding to activate p53[2], was employed to activate p53 in mice. TBI greatly induced the

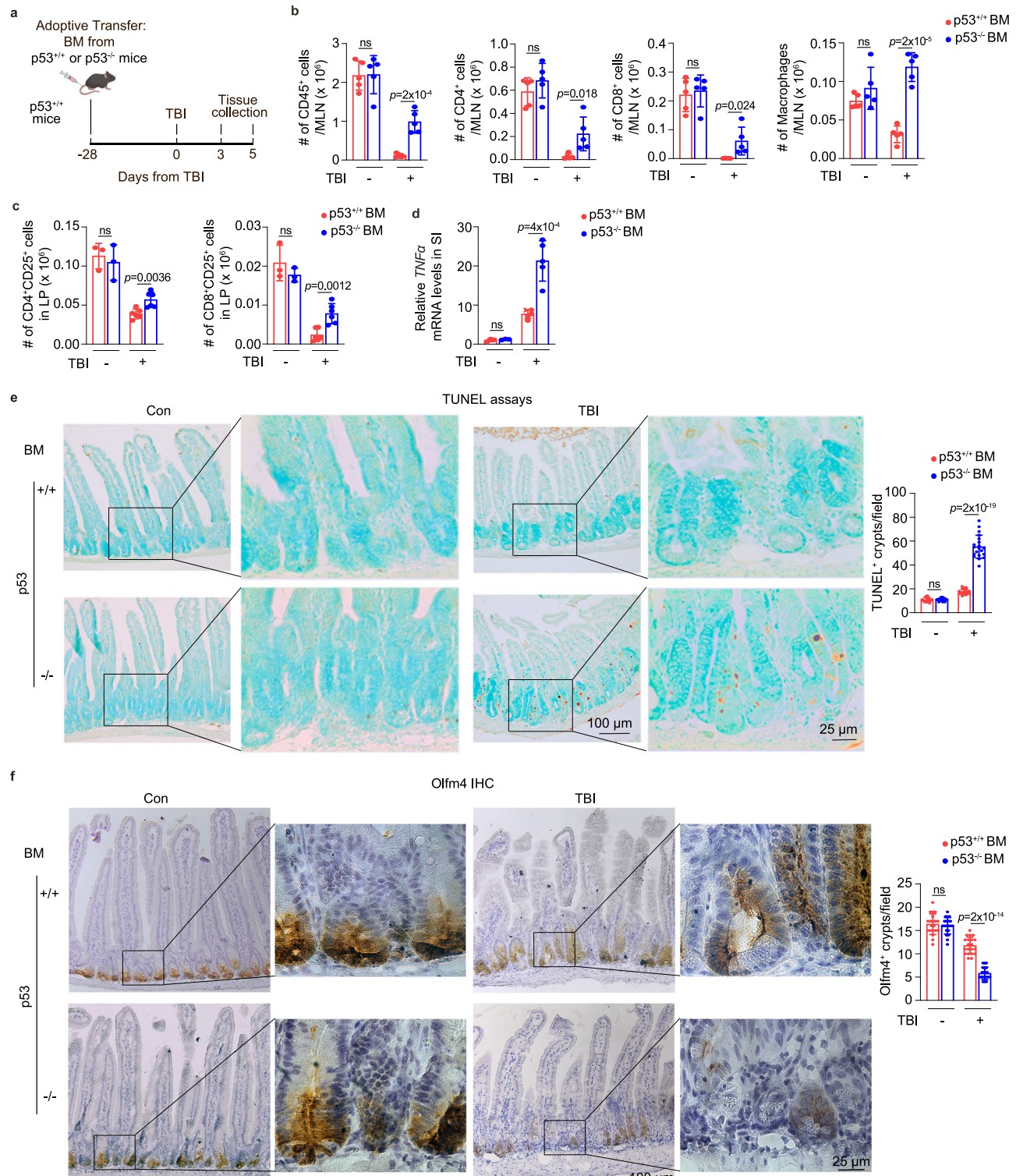

**Fig. 4 | Adoptive transfer of the BM derived from p53$^{-/-}$ mice into p53$^{+/+}$ mice exacerbates the inflammatory immune response and enhances ISC damage post-TBI. a** Schematic diagram of adoptive transfer experimental procedures. **b–d** Adoptive transfer of the p53$^{-/-}$ BM increased the inflammatory immune response post 12 Gy TBI. **b** The number of CD45$^+$, CD4$^+$, CD8$^+$ cells and macrophages in MLNs of p53$^{+/+}$ mice receiving p53$^{+/+}$ or p53$^{-/-}$ BM at 3 days post-TBI as determined by flow cytometric assays. $n = 5$ mice/group. **c** The number of CD25$^+$ activated CD4$^+$ (left) and CD8$^+$ (right) T cells in LP of p53$^{+/+}$ mice receiving p53$^{+/+}$ or p53$^{-/-}$ BM at 5 days post-TBI as determined by flow cytometric assays. $n = 6$ and 3 mice/group for groups with and without TBI treatment, respectively. **d** The mRNA

levels of *TNFα* normalized with the *actin* gene in the SI of p53$^{+/+}$ mice receiving p53$^{+/+}$ or p53$^{-/-}$ BM at 5 days post-TBI as determined by qPCR assays. $n = 5$ and 3 mice/group for groups with and without TBI treatment, respectively. The *TNFα* mRNA levels in mice receiving p53$^{+/+}$ BM without TBI are designated as 1. **e, f** Adoptive transfer of p53$^{-/-}$ BM led to more severe ISC damage post-TBI. Representative images from at least 3 independent mice (left) and quantification (right) of TUNEL (**e**) and IHC staining of Olfm4 (**f**) in the SI of p53$^{+/+}$ mice receiving p53$^{+/+}$ or p53$^{-/-}$ BM at 5 days post-TBI. $n = 20$ fields from at least 3 mice/group. Data are presented as mean ± SD from at least three independent experiments. ns: not significant, two-tailed Student's *t*-test. Source data are provided as a Source Data file.

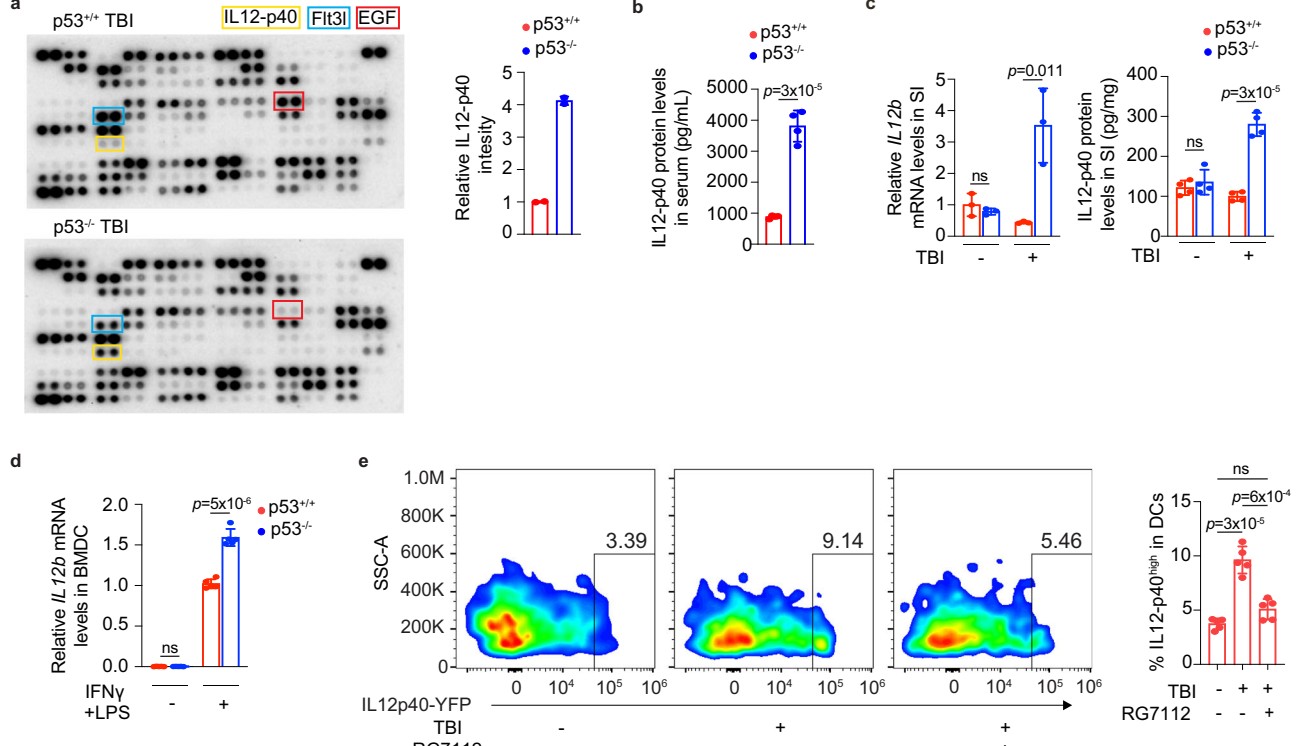

**Fig. 5 | p53 deficiency enhances IL12-p40 expression in mice post-TBI. a** Left: images showing cytokine levels in mouse serum at 3 days post 12 Gy TBI determined by Proteome Profiler Mouse XL Cytokine Array. Right: quantification of serum IL12-p40 levels. Each array represents a pool of 3 mice with the same treatment. Dots in yellow, blue, and red boxes represent IL12-p40, Flt3l and EGF, respectively. **b** The protein levels of IL12-p40 in mouse serum at 3 days post-TBI determined by ELISA assays. $n = 4$ mice/group. **c** The mRNA levels of *IL12b* normalized with the *actin* gene (left) and protein levels of IL12-p40 (right) in the SI of mice at 3 days post-TBI determined by qPCR and ELISA assays, respectively. $n = 3$ and 4 mice/group for left and right panels, respectively. The *IL12b* mRNA levels in p53[+/+] mice without TBI are designated as 1. **d** The mRNA levels of *IL12b* were determined in BMDCs derived from p53[+/+] and p53[−/−] mice stimulated by IFNγ (10 ng/mL) and LPS (100 ng/mL) for 6 h. $n = 5$/group. The *IL12b* mRNA levels in p53[+/+] BMDCs with IFNγ and LPS stimulation are designated as 1. **e** p53 activation by RG7112 reduced the induction of IL12-p40 by TBI. Representative flow cytometric images from at least 3 independent mice (left) and quantification (right) showing the percentage of IL12-p40-YFP[high] DCs in MLNs at 6 h post-TBI in IL12-p40-YFP C57BL/6 reporter mice. $n = 5$ mice/group. Data are presented as mean ± SD from at least 3 independent experiments. ns: not significant, two-tailed Student's *t*-test for (**b**–**d**) and two-tailed Student's *t*-test followed by Bonferroni correction for (**e**). Source data are provided as a Source Data file.

expression of IL12-p40 in DCs from MLNs of IL12p40-YFP mice, as demonstrated by the increased percentage of YFP[high] DCs in MLNs (Fig. 5e). Notably, the induction of IL12-p40 in DCs was greatly reduced upon p53 activation by RG7112 administration (Fig. 5e). The activation of p53 by RG7112 was confirmed by qPCR assays showing the significant induction of the mRNA levels of several well-known p53 target genes, including *Bax*, *PUMA* and *p21*[33,34], in the SI post-TBI and RG7112 treatment (Fig. S11a). These results collectively demonstrate that p53 activation inhibits IL12-p40 expression in DCs of MLNs post-TBI.

It has been reported that IL12 sensitizes the GI tract to irradiation, and administration of recombinant IL12 protein in mice exacerbates radiation-induced GI syndrome and lethality post-TBI[35]. In the context of GVHD, the expression of IL12-p40, which is induced post-TBI, activates donor T cells post-BMT, leading to an inflammatory immune response that triggers the ISC damage and initiates GVHD[7,20]. Here, we investigated whether the regulation of IL12-p40 by p53 contributes to the protective role of p53 in the inflammatory response and GI syndrome induced by TBI. Blocking the function of IL12-p40 by using an aIL12-p40 antibody greatly blocked the inflammatory response in the SI of p53[−/−] mice at 3 days post-TBI, which was reflected by the reduced MHC-II levels on IECs, the reduced number of activated T cells in the LP, and the reduced *TNFα* levels in the SI post-TBI (Figs. 6a–d and S11b). In contrast, aIL12-p40 treatment did not show an obvious effect in p53[+/+] mice (Figs. 6a–d and S11b). In line with the reduced inflammatory immune response, aIL12-p40 treatment in p53[−/−] mice

significantly reduced the number of apoptotic cells in the SI analyzed by the TUNEL assays, significantly reduced the ISC damage reflected by increased numbers of viable Olfm4[+] ISCs and lysozyme[+] Paneth cells in the SI at 3 days post-TBI, and most importantly, significantly increased mouse lifespan post-TBI (Figs. 6e–g and S11c). These results suggest a crucial mechanism of p53 in regulating T cell activation through inhibition of the IL12-p40 expression, which in turn reduces the ISC damage and protects against radiation-induced GI toxicity.

## p53 modulates IL12-p40 levels post-TBI through LIF

To identify the downstream mediator(s) involved in the protective role of p53 in the inflammatory response to TBI in the SI, the expression of a panel of p53 target genes in the SI of p53[+/+] and p53[−/−] mice at 1 day post-TBI was compared using the data from RNA-seq analysis described in Fig. 1b. p53 deficiency led to the reduced expression of several well-known p53 target genes that are involved in apoptosis (*Bax* and *PUMA*) and repair (*p21*)[33,34], which may contribute to the loss of early apoptosis and the attenuated repair in the SI of p53[−/−] mice post-TBI (Fig. 7a). While the products of these p53 target genes contribute to some of the well-characterized p53 functions, such as apoptosis, no report has suggested their involvement in regulating IL-12 production by DCs. Interestingly, the levels of *LIF*, an important cytokine and known p53 target, were significantly lower in the SI of p53[−/−] mice compared with p53[+/+] mice post-TBI (Fig. 7a). Previously, we reported that p53 transcriptionally induces LIF expression, which mediates the

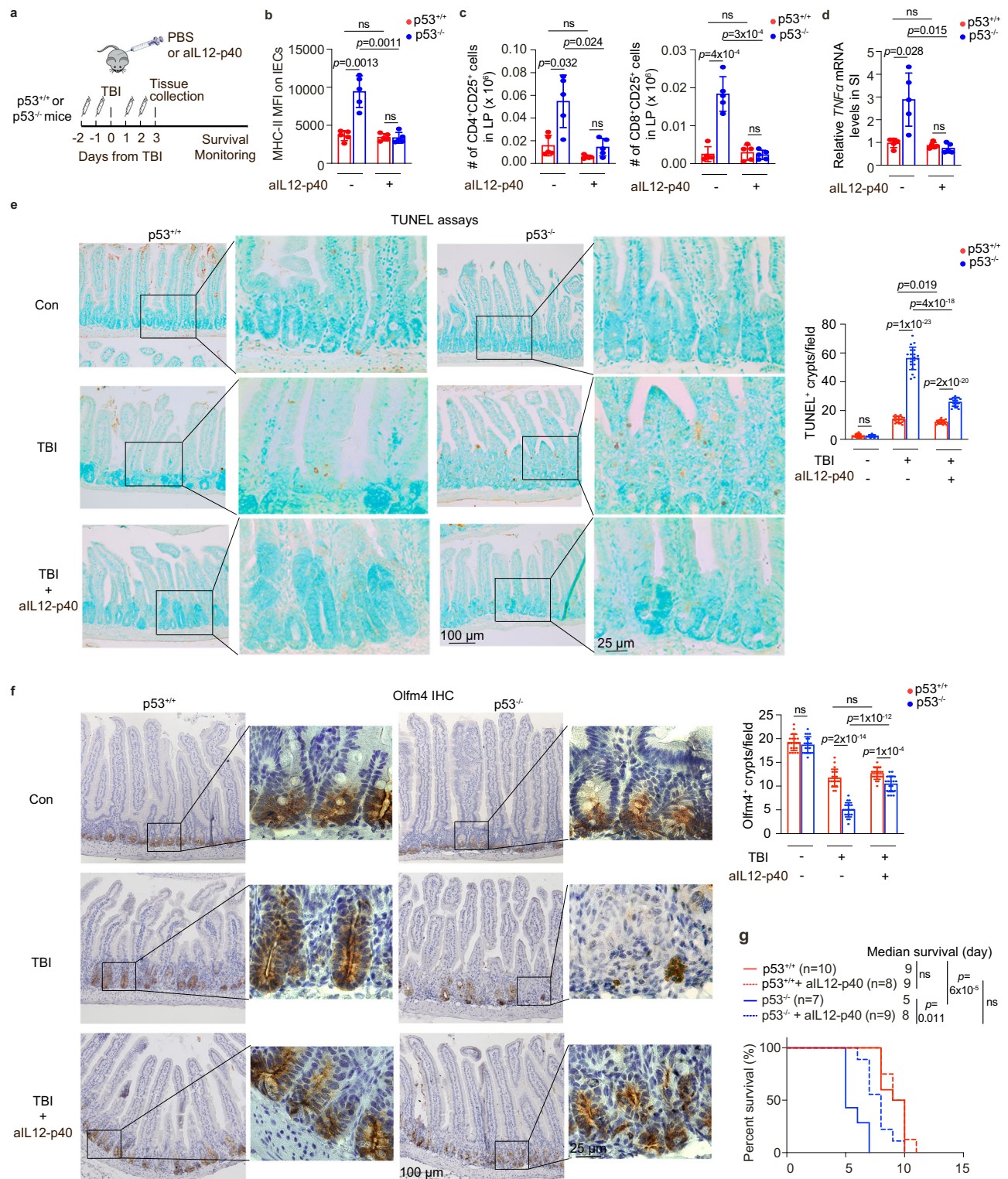

**Fig. 6 | p53 protects mice from radiation-induced damage by inhibiting IL12-p40 expression. a** Schematic diagram of experimental procedures of aIL12-p40 antibody administration (i.p., 200 μg/mouse, once a day for 4 days from 2 days before till 2 days post-TBI). **b–g** aIL12-p40 antibody preferentially inhibited the inflammatory immune response in p53⁻/⁻ mice post-TBI. **b** MFI of MHC-II on IECs from mice at 3 days post-TBI. *n* = 5 mice/group. **c** The number of CD25⁺ activated CD4⁺ (left) and CD8⁺ (right) T cells in LPs of mice at 3 days post-TBI. *n* = 5 mice/group. **d** The mRNA levels of *TNFα* normalized with the *actin* gene in the SI of mice at 3 days post-TBI. *n* = 5 mice/group. The *TNFα* mRNA levels in p53⁺/⁺ mice post-TBI are designated as 1. **e**, **f** aIL12-p40 antibody protected ISCs from TBI-induced

damage. Representative images from at least 3 independent mice (left) and quantification (right) of TUNEL (**e**) and IHC staining of Olfm4 (**f**) in the SI of mice with or without aIL12-p40 antibody administration at 3 days post-TBI. *n* = 20 fields from at least 3 mice/group. **g** Kaplan-Meier survival curves of mice treated with or without aIL12-p40 antibody post 12 Gy TBI. For (**b**–**f**) data are presented as mean ± SD from at least 3 independent experiments. ns: not significant, two-tailed Student's *t*-test followed by Bonferroni correction for (**b**–**f**) and Kaplan-Meier survival analysis followed by Bonferroni correction for (**g**). Source data are provided as a Source Data file.

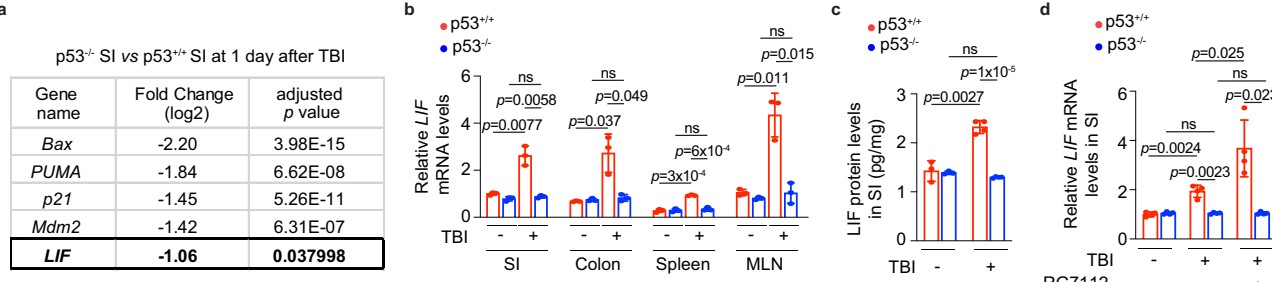

**Fig. 7 | p53 induces LIF expression post-TBI. a** Differential expression of a panel of p53 target genes in the SI of p53⁻/⁻ mice versus p53⁺/⁺ mice at 1 day post 12 Gy TBI. **b**, **c** The mRNA levels of *LIF* normalized with the *actin* gene in the SI, colon, spleen, and MLN (**b**) and the protein levels of LIF in the SI (**c**) of p53⁺/⁺ and p53⁻/⁻ mice at 3 days post 12 Gy TBI determined by qPCR and ELISA assays, respectively. For (**b**) *n* = 3 mice/group. For (**c**) *n* = 4 and 3 mice/group for groups with and without TBI treatment, respectively. The *LIF* mRNA levels in the SI of p53⁺/⁺ mice without TBI are designated as 1. **d** p53 activation by RG7112 increased the mRNA levels of *LIF* in the SI of p53⁺/⁺ mice at 6 h post 12 Gy TBI. *n* = 4 mice/group. The *LIF* mRNA levels in p53⁺/⁺ BMDCs without stimulation are designated as 1. Data are presented as mean ± SD from at least 3 independent experiments. ns: not significant, two-tailed Student's *t*-test followed by Bonferroni correction. Source data are provided as a Source Data file.

function of p53 in embryonic implantation[36]. Recently, we found that LIF is essential in maintaining ISC number and functions to ensure the intestinal epithelial homeostasis and regeneration[4], and protects against GVHD by inhibiting donor T cell-mediated inflammatory immune response post allogenic BMT[20]. These studies suggest that LIF may function as an important mediator for the protective role of p53 in the inflammatory response to TBI in the SI. The lower *LIF* expression in p53⁻/⁻ mice was validated in different tissues; TBI significantly induced the mRNA expression of *LIF* in multiple tissues, including the SI, colon, spleen and MLN, in p53⁺/⁺ but not p53⁻/⁻ mice (Fig. 7b). p53-dependent induction of LIF was also confirmed at the protein levels in the SI as measured by ELISA assays (Fig. 7c). Activating p53 using RG7112 further enhanced the expression of *LIF* induced by TBI in the SI of p53⁺/⁺ mice but not p53⁻/⁻ mice (Fig. 7d), supporting the p53-dependent induction of LIF by TBI in the SI.

We next investigated whether LIF mediates the inhibitory effect of p53 on inflammation and the protective role in ISCs post-TBI. rLIF (i.p. 30 ng/g body weight, twice/day) was administrated to both p53⁺/⁺ and p53⁻/⁻ mice for 7 days, starting 3 days before TBI till 3 days after TBI (Fig. 8a). rLIF administration largely abolished the elevated IL12-p40 protein levels in p53⁻/⁻ mouse serum at 3 days post-TBI as determined by using the cytokine panel (Fig. S12) and validated in the SI and serum by ELISA assays (Fig. 8b). We also observed an inhibitory effect of rLIF on IL12-p40 production in p53⁺/⁺ mice, but to a much less extent than that in p53⁻/⁻ mice (Fig. 8b). Consistent with the increased *LIF* expression in tissues in p53⁺/⁺ mice post-TBI (Fig. 7b), stimulation of IFNγ and LPS increased *LIF* expression in p53⁺/⁺ BMDCs but not p53⁻/⁻ BMDCs (Fig. 8c). Given the higher induction of *IL12b* and lower *LIF* levels in p53⁻/⁻ BMDCs stimulated by IFNγ and LPS compared with p53⁺/⁺ BMDCs, we investigated whether the lower LIF levels in p53⁻/⁻ BMDCs contribute to the higher production of *IL12b*. rLIF treatment in BMDCs stimulated by IFNγ and LPS reduced the *IL12b* expression in BMDCs, especially in p53⁻/⁻ BMDCs (Fig. 8c), suggesting that LIF mediates the regulation of IL12-p40 by p53 in DCs. In line with the reduced IL12-p40 expression in p53⁻/⁻ mice upon rLIF treatment, rLIF largely abolished the increased inflammation in p53⁻/⁻ mice at 3 days post-TBI; rLIF reduced MHC-II expression on IECs, reduced the number of activated CD4⁺ and CD8⁺ T cells in the LP, and inhibited *TNFα* expression in the SI of p53⁻/⁻ mice post-TBI (Fig. 8d–f). Notably, the same treatment exhibited a very limited effect on p53⁺/⁺ mice (Fig. 8d–f).

Our recent study revealed a crucial role of LIF in regulating ISC functions[4,20]. We observed that compared with p53⁺/⁺ intestinal organoids, p53⁻/⁻ intestinal organoids showed significantly reduced levels of *LIF* expression (Fig. S13), which may account for the increased vulnerability of p53⁻/⁻ intestinal organoids towards T cell-induced damage

(Fig. 3). Here, we examined the effect of rLIF administration on activated T cell-induced cell death in p53⁺/⁺ and p53⁻/⁻ organoids. While rLIF treatment in p53⁺/⁺ organoids co-cultured with activated T cells significantly reduced T-cell triggered cell death, rLIF treatment displayed a much more pronounced effect on p53⁻/⁻ organoids (Fig. 8g), strongly suggesting that the regulation of LIF expression by p53 mediates the sensitivity of intestinal organoids towards activated T cell-induced damage. Given the critical role of LIF in reducing inflammatory response and protecting ISCs, we examined the effect of rLIF administration on ISCs in p53⁺/⁺ and p53⁻/⁻ mice post damage. rLIF administration significantly ameliorated TBI-induced ISC damage in both p53⁺/⁺ and p53⁻/⁻ mice; rLIF significantly reduced the number of TUNEL⁺ apoptotic cells, and increased the number of viable Olfm4⁺ ISCs and lysozyme⁺ Paneth cells in the SI of both p53⁺/⁺ and p53⁻/⁻ mice at 3 days post-TBI. Notably, the protective effect of rLIF on ISCs was more pronounced in p53⁻/⁻ mice (Figs. 8h, S14 and S15). In turn, while rLIF prolonged the survival of both p53⁺/⁺ and p53⁻/⁻ mice, rLIF exhibited a much more pronounced effect on the survival of p53⁻/⁻ mice post-TBI than p53⁺/⁺ mice, suggesting that LIF mediates the protecting effect of p53 on TBI-induced GI syndrome (Fig. 8i). Taken together, these results indicate that the p53-mediated induction of LIF is essential for p53 to repress IL12-p40 expression in DCs, suppress the inflammatory response, and ameliorate the ISC damage post-TBI, which in turn contribute to the protective role of p53 in TBI-induced GI syndrome.

### Transient rLIF administration has no obvious effect on intestinal tumor growth in Apc^Min/+ mice

In addition to minimizing the damage in normal tissues, another essential aspect of a successful treatment against radiation-induced GI syndrome is the safety of treatments. LIF overexpression in solid tumors has been reported to be associated with poor clinical outcomes in patients, which is supported by the observations that constant (stable) LIF overexpression in cancer cells promotes tumor progression in animal models[37,38]. Here, we tested whether rLIF treatment for a short period of time will affect tumor development by employing the Apc^Min/+ mice that contain a mutation in the tumor suppressor *Apc* gene and are prone to develop spontaneous intestinal tumors[39,40]. rLIF was administrated to Apc^Min/+ mice for 7 days, a regimen that protects against radiation-induced GI damage (Fig. 8), at the age of 60 days, when intestinal tumors start to occur, or at the age of 90 days, when intestinal tumors are developing[40] (Fig. 9a). rLIF treatment had no obvious effect on the number and size of intestinal tumors in these two groups of mice (Fig. 9a), suggesting that transient rLIF administration does not promote tumorigenesis. We further validated that this

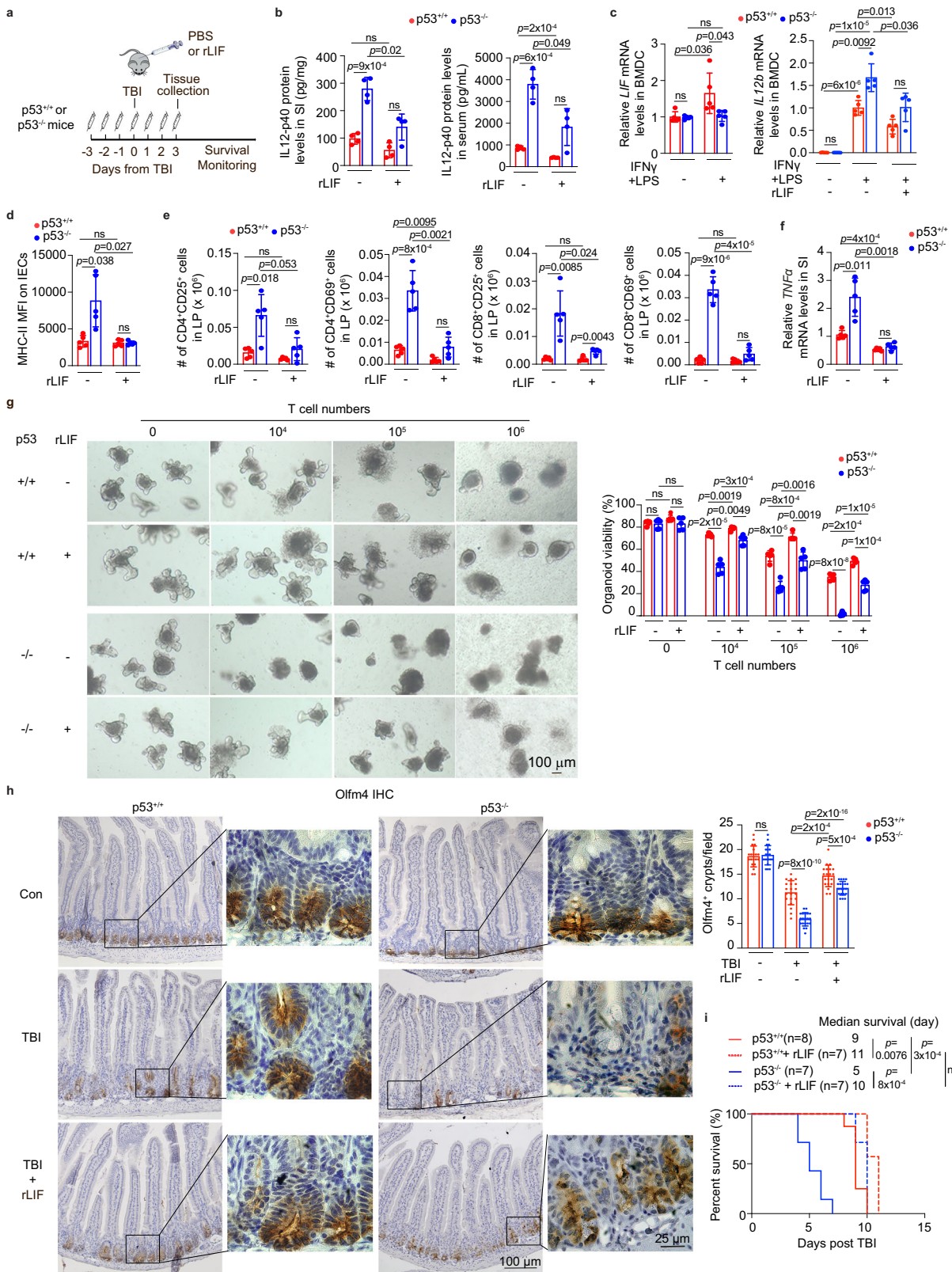

treatment of rLIF exhibited a protective effect on radiation-induced GI damage in Apc^Min/+ mice; rLIF treatment significantly promoted the lifespan of Apc^Min/+ mice subjected to 12 Gy TBI at the age of 120 days, when intestinal tumors are fully developed (Fig. 9b). These data highlight the potential application of rLIF in preventing radiation-induced GI syndrome in cancer patients (Fig. 9c).

## Discussion

As the "guardian of the genome", p53 is most well-known for its ability to sustain genome integrity and suppress tumor growth[9,41]. In response to different extra- and intra-cellular stimuli, p53 stabilizes and induces different sets of target genes involved in many different biological functions in a highly cell-, stress-, and environment-dependent

**Fig. 8 | rLIF administration preferentially inhibits the inflammatory immune response and protects ISCs in p53[-/-] mice post-TBI. a** Schematic diagram of the experimental procedure. **b** The protein levels of IL12-p40 in the SI (left) and serum (right) of mice at 3 days post-TBI determined by ELISA assays. *n* = 4 mice/group. **c** The mRNA levels of *LIF* (left) and *IL12b* (right) normalized with the *actin* gene in p53[+/+] and p53[-/-] BMDCs with or without IFNγ and LPS stimulation. *n* = 5/group. The *LIF* mRNA levels in p53[+/+] BMDCs without stimulation are designated as 1. The *IL12b* mRNA levels in p53[+/+] BMDCs with stimulation are designated as 1. **d-f** rLIF inhibited the elevated inflammatory immune response in p53[-/-] mice post-TBI. **d** MFI of MHC-II on IECs from mice at 3 days post-TBI. *n* = 5 mice/group. **e** The number of CD25[+] or CD69[+] activated CD4[+] (left panels) and CD8[+] (right panels) T cells in the LP of mice at 3 days post-TBI. *n* = 5 mice/group. **f** The mRNA levels of *TNFα* normalized with the *actin* gene in the SI of mice at 3 days post-TBI. *n* = 5 mice/group. **g** The effect of rLIF

on intestinal organoid susceptibility towards T cell-induced damage. Left: representative images of organoid growth in the presence of activated p53[+/+] T cells; right: quantification of organoid viability measured at 2 days post co-culture using the CyQUANT™ LDH Cytotoxicity Assay Kit. *n* = 5/group. **h** rLIF protected ISCs from TBI-induced damage. Representative images (left) and quantification (right) of IHC staining of Olfm4 in the SI of mice with or without rLIF administration at 3 days post-TBI. *n* = 20 fields from at least 3 mice/group. **i** Kaplan–Meier post-TBI survival curves of mice treated with or without rLIF. For (**b–h**) data are presented as mean ± SD from at least 3 independent experiments. ns: not significant, Kaplan-Meier survival analysis followed by Bonferroni correction for (**i**) and two-tailed Student's *t*-test followed by Bonferroni correction for other panels. Source data are provided as a Source Data file.

manner[9,41,42]. While many of these functions are related to its tumor suppressive activities, some of them also contribute to additional important p53 functions. Emerging evidence has suggested an essential role of p53 in regulating immunity and inflammatory responses[42–44]. For example, upon viral infection, p53 transcriptionally induces its target genes, including IRF9 and TLR3, to activate a type I IFN antiviral immune response[44]. Our recent study showed that p53 induces type 2 innate immunity through transcriptional induction of LRMP to protect against parasitic infections[42]. Further, p53 inhibits infection- and injury-triggered chronic inflammation, which in turn suppresses the chronic inflammation-induced tumorigenesis[43]. These studies have demonstrated a broader role of p53 in maintaining the homeostasis of the immune system.

The role of p53 in normal tissues in response to radiation is complex[2,11]. In the hematopoietic system, the presence of p53 triggers radiation-induced apoptosis to promote the hematopoietic injury; loss of p53 in mice blocks radiation-induced apoptosis and ameliorates hematopoietic toxicity induced by radiation[11]. On the contrary, in the GI tract, p53 exerts a protective role by reducing GI damage and limiting radiation-induced GI toxicity. IECs are highly sensitive to radiation-induced apoptosis which is p53-dependent and occurs shortly after irradiation. However, while IECs in p53[-/-] mice are resistant to radiation-induced apoptosis shortly after irradiation, p53[-/-] mice exhibit a more severe GI syndrome, indicating that additional functions of p53 other than apoptosis in IECs are involved in GI's response towards radiation[2,11,12]. A previous study using an MDM2 mutant mouse model that has a disrupted p53-MDM2 feedback loop and enhanced p53 activity in response to DNA damage showed that p53 activation protects the mice from GI toxicity[2]. Consistently, enhancing p53 activity in wild type (WT) mice using small molecule RG7112 promotes intestinal epithelial recovery, and prolongs mouse lifespan post-irradiation[2]. Although these studies have suggested a protective role of p53 in radiation-induced GI toxicity, its precise underlying mechanism is far from clear.

This study demonstrates that p53 protects against radiation-induced ISC damage by limiting the antigen-presenting function of IECs to inhibit the inflammation triggered by TBI, which in turn mitigates GI syndrome and prolongs mouse lifespan. In the GI tract, while radiation activates p53 to induce apoptosis in IECs and ISCs that causes blunting of the villus immediately after irradiation, p53-mediated apoptosis also eliminates a large number of immune cells. Further, in response to radiation, p53 suppresses IL12 production from DCs in MLNs and the GI tract. Recent studies, including ours, have shown that the elevation of IL12 increases MHC-II expression on IECs, which serve as APCs to activate T cells to induce GI injury[7,20]. Compared with p53[+/+] mice, p53[-/-] mice have a much higher induction of IL12-p40 post-TBI, which in turn leads to much higher levels of MHC-II on IECs. Elevated MHC-II on IECs induces CD4[+] T cell activation, which can further activate CD8[+] T cells through direct interaction and cytokine secretion[45,46]. Both activated CD4[+] and CD8[+] T cells secrete TNFα to enhance intestinal inflammation and ISC damage, which lead to more severe

radiation-induced GI toxicity in p53[-/-] mice. Importantly, administering the aIL12-p40 antibody preferentially inhibits the inflammation and prolongs the survival of p53[-/-] mice post-TBI. Mechanistically, p53 inhibits IL12-p40 production through transcription induction of LIF. These results demonstrate that the p53 regulation of LIF/IL12/MHC-II axis and the p53-mediated apoptosis that eliminates immune cells play an important role in producing an immunosuppressive environment in the gut post-TBI to protect against radiation-induced GI toxicity (Fig. 9c). It has been reported previously that in some types of tumor cells, such as myeloid leukemia (AML) and melanoma cancer cells, p53 activation increases the transcription of MHC-II to increase anti-cancer immune response[47,48]. Further, in the context of melanoma, p53 enhances the production of IL12 in APCs to promote anti-cancer cytotoxicity[49]. The dual effects of p53 on MHC-II expression in tumor cells and IECs, and IL12 production in APCs reflect that p53 exerts its function in a highly context-dependent manner which leads to increased anti-cancer immune response towards cancer cells and reduced inflammation in the GI tract to limit radiation-induced GI toxicity, respectively. It's worth noting that p53[-/-] intestinal organoids show significantly reduced *LIF* expression which is associated with increased vulnerability to T cell-induced damage, and rLIF treatment displays a much more pronounced effect on reducing T cell-triggered cell death in p53[-/-] organoids than in p53[+/+] organoids when co-cultured with activated T cells. These results suggest that in addition to down-regulating IL12 production in DCs and thus MHC-II expression on IECs, p53 directly protects ISCs from activated T cell-induced damage through its transcriptional regulation of LIF (Fig. 9c). Administering rLIF preferentially inhibits the inflammation, reduces the ISC damage, and prolongs the survival of p53[-/-] mice post-TBI. Compared with the effect of an aIL12-p40 antibody, rLIF treatment shows a more pronounced rescuing effect on radiation-induced GI toxicity in p53[-/-] mice.

The intestinal epithelium shows remarkable flexibility upon damage. A variety of additional cell types in addition to Lgr5[+] ISCs can acquire stem cell characteristics after injury and are involved in damage-induced epithelial regeneration. For example, quiescent stem cells, which reside in the +4- +6 cell position from the crypt base, are resistant to stress and can repopulate the crypt upon injury, functioning as reserve ISCs[50]. In addition, several lineage-committed progenitor cells, including ATOH1[+] or DLL-1[+] secretory progenitors or ALPI[+] enterocyte progenitors, Paneth cells, Enteroendocrine cells and Tuft cells have all been shown to be able to acquire stemness property and contribute to injury-induced regeneration[51–56]. For many of the cell types discussed above, the reversion to an Lgr5[+] ISC state is required for their contribution to the intestinal epithelial regeneration[57,58]. Future studies are needed to characterize the effect of p53 on the regenerative activity of these different cell types located within and outside of the crypt upon injury.

Taken together, our study reveals a crucial function of p53 in inhibiting the antigen-presenting function of IECs to ameliorate inflammation upon TBI, which in turn protects against radiation-

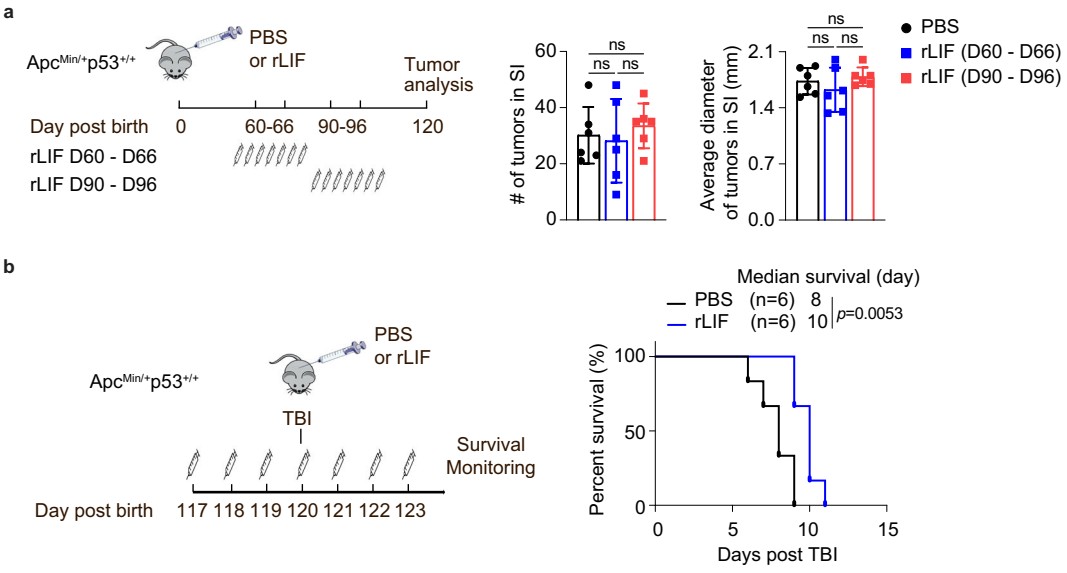

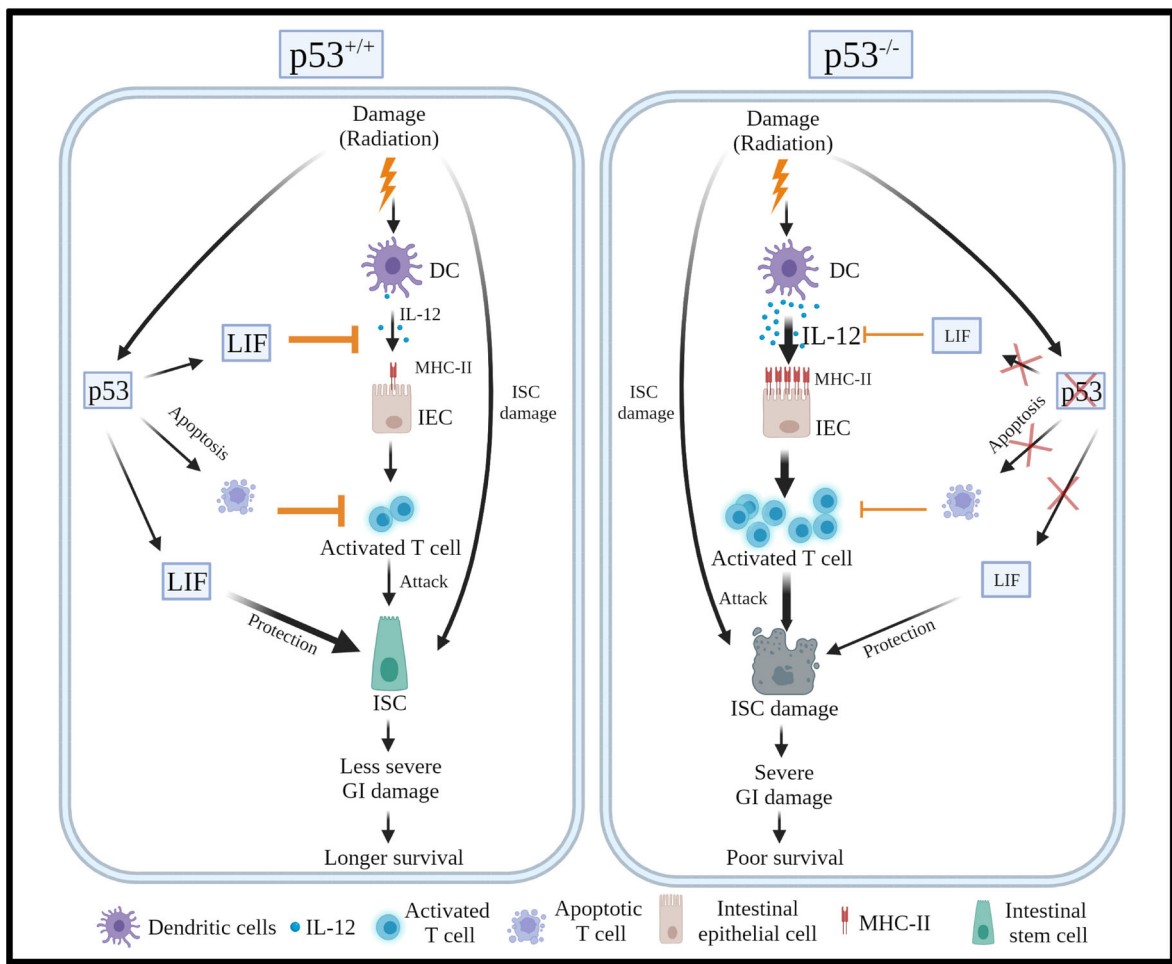

**Fig. 9 | Transient rLIF administration has no obvious effect on intestinal tumor growth in mice. a** Transient rLIF administration showed no obvious effect on intestinal tumor growth in Apc^Min/+p53^+/+ mice. Left: schematic diagram of experimental procedures. Right: quantification of tumor number and tumor diameter in the SI of Apc^Min/+p53^+/+ mice at the age of 120 days with or without rLIF administration. $n = 6$ mice/group. Data are presented as mean ± SD from at least 3 independent experiments. ns: not significant, two-tailed Student's *t*-test followed by Bonferroni correction. **b** Transient rLIF administration prolonged the lifespan in Apc^Min/+p53^+/+ mice post-TBI. Left: schematic diagram of experimental procedures (rLIF administration: i.p., 30 ng/g body weight, twice a day for 7 days from 3 days before till 3 days post 12 Gy TBI). Right: Kaplan-Meier post-TBI survival curves of Apc^Min/+p53^+/+ mice treated with or without rLIF. **c** Schematic illustration of the role of p53 in protecting against radiation-induced GI syndrome. The diagram was created with BioRender.com. Source data are provided as a Source Data file.

induced ISC damage and GI syndrome. The p53-mediated apoptosis of immune cells, the regulation of the LIF/IL12/MHC-II axis to inhibit T cell activation, and the protecting effect of LIF on ISC are important underlying mechanisms by which p53 protects against radiation-induced GI syndrome, which has potential translational applications to protect against radiation-induced GI syndrome.

## Methods

### Mice

All mouse experiments were approved by the Institutional Animal Care and Use Committee (IACUC) of Rutgers University. WT p53 (p53[+/+], Cata#: 000664), p53-deficient (p53[−/−], Cata#: 002101), CD45.1 (Cata#: 002014) and Apc[Min/+] (Cata#: 002020) mice were obtained from the Jackson Laboratory. IL12-p40-IRES-eYFP mice were a kind gift from Dr. Timothy E. O'Sullivan at UCLA. Mice were housed under a 12-h light/dark cycle with 6 am light on and 6 pm light off. The temperature was maintained between 70° and 74 °F and the humidity was between 30 and 70%. Age- and gender-matched mice at 8–12 weeks old were used for experiments in this study. Animals were randomly assigned to different treatment groups. Sample sizes were chosen based on the power calculation. For TBI treatment, mice were subjected to 12 Gy TBI using a Cs-137 γ-source irradiator at a dose rate of 90 cGy/min. rLIF (Cata#: ESG1107, Millipore), aIL12-p40 antibody (Cata#: 505309, Biolegend), aCD3 antibody (Cata#: BE0001-1, Bio X Cell), IgG (Cata#: BE0091, Bio X Cell), and RG7112 (Cata#: S7030, Selleck Chemicals) were used for mouse treatments. The investigators were blinded to the group allocation during experiments and when assessing outcomes.

### H&E staining assays

H&E staining of formalin-fixed, paraffin-embedded (FFPE) sections was performed as previously described[4]. Briefly, tissue sections were deparaffinized in xylene and rehydrated in ethanol and water, followed by staining with hematoxylin and eosin.

### IHC staining assays

IHC staining of FFPE sections was performed as previously described[4]. Briefly, tissue sections were deparaffinized in xylene and rehydrated in ethanol and water, followed by antigen retrieval by boiling slides in antigen unmasking solution (Cata#: h3300, Vector Laboratories) for 10 min. Immunostaining was performed using an anti-Olfm4 antibody (Cata#: 39141 S, Cell Signaling, 1:1000 dilution) overnight at 4 °C, followed by incubating with the biotinylated goat anti-rabbit IgG (H + L) antibody (Cata#: BA-1000-1.5, Vector Laboratories, 1:200 dilution).

### TUNEL staining assays

TUNEL staining was performed by using a TUNEL assay kit (Cata#: ab206386, Abcam) according to the manufacturer's instructions.

### IF staining assays

IF staining of FFPE sections was performed as previously described[4]. Briefly, tissue sections were deparaffinized in xylene and rehydrated in ethanol and water, followed by antigen retrieval by boiling slides in antigen unmasking solution (Cata#: h3300, Vector Laboratories) for 10 min. After pre-incubation with 2% BSA and 2% goat serum, tissue sections were incubated with the anti-lysozyme (Cata#: ab108508, Abcam, 1:5000 dilution), anti-Olfm4 (Cata#: 39141 S, Cell Signaling, 1:1000 dilution), anti-CD45 (Cata#: 550539, BD, 1:200 dilution) or anti-CD3 (Cata#: 99940, Cell Signaling, 1:200 dilution) antibodies overnight at 4 °C, followed by incubating with the goat anti-rabbit IgG (H + L) Cy5® (Cata#: ab6564, Abcam, 1:200 dilution), goat anti-rabbit IgG (H + L) Alexa Fluor™ 555 (Cata#: A-21428, Invitrogen, 1:200 dilution) or goat anti-rat IgG (H + L) Alexa Fluor™ 555 (Cata#: ab150158, Abcam, 1:200 dilution) antibodies. Nuclei were stained with 4′, 6-diamidino-2-phenylindole (DAPI; Vector Labs). IF staining of frozen sections for MHC-II was performed as previously described[20]. Anti-MHC-II- Alexa Fluor® 594 (Cata#: 107650, Biolegend, 1:200 dilution) was used. Images were acquired using a Nikon A1R-Si Confocal Microscope System.

### Flow cytometric assays

The spleen, MLN, LP and IECs from the SI were collected and single-cell suspensions were prepared for flow cytometric analysis as previously described[20]. Gating strategies are shown in Fig. S5. The antibodies used for flow cytometric assays are listed in Supplementary Table S1. Samples were collected using an Attune NxT Flow Cytometer (Thermo-Fisher) and analyzed by FlowJo 10 software (Tree Star).

### RNA extraction and qPCR assays

The total RNA extraction and reverse transcription to cDNA were performed as previously described[4]. The Taqman primers for mouse *LIF* (Cata#: Mm00434761_m1), mouse *IL12b* (Cata#: Mm01288989_m1), and mouse *actin* (Cata#: 4352933E) were purchased from Applied biosystems. The Taqman assays were performed using Taqman™ Gene Expression Master Mix (Cata#: 4369016, Applied biosystems). The expression levels of *TNFα*, *BAX*, *PUMA* and *p21* were determined by the SYBR Green assays. Sequences of the SYBR Green primers are listed in Supplementary Table S2. The SYBR Green assays were performed using PowerUp™ SYBR Green Master Mix (Cata#: A25778, Applied biosystems). The mRNA levels of the analyzed genes were normalized with the *actin* gene.

### RNA-seq analysis

The total RNA of the SI was extracted using RNeasy Mini kits (Cata#: 74106, Qiagen). RNA-seq sequencing and analysis were performed by Novogene Co., Ltd.

### ELISA assays

The protein levels of LIF and IL12-p40 were determined using a LIF Duoset kit (Cata#: DY449, R&D) and an IL12-p40 Duoset kit (Cata#: DY2398-05, R&D), respectively.

### Cytokine panel assays

The cytokine levels in the mouse serum were measured using the Proteome Profiler Mouse XL Cytokine Array (Cata#: ARY028, R&D) following the manufacturer's instructions.

### BMDC culture

BMDC culture was performed as previously described[20]. BMDCs were cultured for 6 days before stimulation and harvesting. Mouse rLIF (Cata#: ESG1107, Millipore, 100 ng/mL), IFNγ (Cata#: I4777, Sigma, 10 ng/mL) and LPS (Cata#: L4391, Sigma, 100 ng/mL) were used to treat BMDCs.

### Adoptive transfer

Transplant recipients were lethally irradiated with 11 Gy TBI (split doses of 2 × 5.5 Gy with a 4-h interval), followed by intravenous (i.v.) transplantation of 5 × 10[6] BM cells at 1 day post-TBI. BM cells were prepared by flushing cells out from intact femurs and tibiae, followed by the removal of red blood cells using the RBC buffer (Cata#: 420302, Biolegend).

### Intestinal organoid culture

Intestinal crypt isolation and organoid culture were performed as previously described[4]. The intestinal organoid and T-cell co-culture was performed as previously described[26]. Briefly, p53[+/+] and p53[−/−] intestinal organoids were cultured for 3 days. On day 3, mouse splenocytes were isolated from p53[+/+] mice, and T cells were enriched using a Pan T cell isolation kit (Cata#: 130-095-130, Miltenyi Biotec), followed by stimulation with PMA (Cata#: P1585, Sigma, 50 ng/mL) and ionomycin (Cata#: I9657, Sigma, 1 μg/mL) for 2 h at 37 °C. Organoids and

stimulated T cells were mixed in a 1.5 mL tube with 1 mL of DMEM. After incubation at 37 °C for 5 min, the mixture was centrifuged for 2 min at 200 × g. The pellet was suspended in 30 μL of Matrigel, and seeded into 24-well plates. The organoid viability was measured at 2 days post co-culture using the CyQUANT™ LDH Cytotoxicity Assay Kit (Cata#: C20301, Thermo Scientific) following the manufacturer's instructions.

### Statistical analysis

All data were obtained from at least three repetitions and were presented as mean ± SD. The survival of mice was summarized by Kaplan-Meier plots and compared by the log-rank (Mantel–Cox) test using the GraphPad Prism software. All other p-values were obtained by two-tailed Student's t-tests. The p-values were adjusted for multiple tests using the Bonferroni correction[59]. Values of $p < 0.05$ were considered significant. All data points and "n" values reflect biological replicates.

### Reporting summary

Further information on research design is available in the Nature Portfolio Reporting Summary linked to this article.

## Data availability

The RNA-seq data generated in this study have been deposited in the Gene Expression Omnibus (GEO) database under accession code GSE226421. Source data are provided with this paper.

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

## Acknowledgements

This work was supported in part by grants from NIH 1R01CA260837 and R01CA260838 to W.H. and grants from NIH 1R01CA227912 and 1R01CA214746 to Z.H. P.X. is supported by the grant from NIH R21AI128264. J.W. was supported by a NJCCR Fellowship Award COCR24PDF006. This work was also supported by the Flow Cytometry/Cell Sorting shared resource of Rutgers Cancer Institute of New Jersey (NIH P30CA072720).

## Author contributions

J.W. carried out the experiments, analyzed data and wrote the manuscript; C.C., X. Y., F. Z., J.L., carried out experiments; L.Z. assisted with histological analysis; P.X. assisted with flow cytometric analysis of immune cell composition and activities; J.B. assisted with assessing p53 function in DNA damage response; Z.F., W.H. designed experiments, analyzed data and wrote the manuscript.

## Competing interests

The authors declare no competing interests.
