## [Peer Review File · Nature Communications]

p53 suppresses MHC class II presentation by intestinal epithelium to protect against radiation-induced gastrointestinal syndromeREVIEWER COMMENTS

Reviewer #1 (Remarks to the Author):

The authors of the submitted manuscript investigate the intestinal axis P53/MHCII/IL-12 during the response to radiation injury. The authors demonstrated that WT mice have better survival rates after total-body irradiation than p53^{-/-} mice. They showed that WT mice response is accompanied by earlier and shorter apoptosis in the crypts and efficient regeneration of the crypt compartment. They showed that p53^{-/-} mice have significantly increased and prolonged inflammatory responses compared to WT mice. Furthermore, they demonstrated that immune cells lacking p53 increase inflammation and p53^{-/-} mice have DC cells expressing IL-12-p40 upon TBI. Utilizing multiple in vivo and in vitro models, they showed that p53 regulates IL-12-p40 expression upon radiation injury. The submitted manuscript is exciting, and I have several comments.

Major.

There have been studies showing p53/IL-12/p40 in gut regeneration that should be discussed and addressed: PMID 27141366.

The authors should improve the quality of immunofluorescence and immunohistochemistry images. In addition, the authors should include insets with a higher magnification.

In several places, the authors refer to the results obtained from the p53^{-/-} mouse model as proof that intestinal stem cells are only responsible for the regenerative capacity of the gut. Now, there have been many papers showing that multiple cells within the crypts participate in the regeneration and can originate from very different cell populations. I recommend the authors address it or rewrite appropriate sections that, at this stage, are a bit far-reaching.

I am unsure whether the authors showed that MHC II on only intestinal epithelial cells participates in signal transduction.

In addition, the results presented in Figure 7A still need to be completed. As irradiation and p53 are significant players in the suggested model, the authors should show whether rLIF plays a role in irradiation.

Reviewer #3 (Remarks to the Author):

In this manuscript, Wang and coworkers study the mechanisms by which ISC is protected from radiotherapy: p53 protects ISC from radiotherapy by inhibiting IL12p40/MHC-II pathway via induction of LIF. While the study is generally well performed and presented, inflammatory cascade of LIF/IL12p40/MHC-II/T cell axis in another GI syndrome model (Wang J. et al., Blood 2022), the protective outcome of LIF on ISC against radiation (Wang H. et al., Cell Death & Disease 2020), and transcriptional induction of LIF by p53 (Hu W. et al, Nature 2007), have been already described by the authors themselves in their previous studies. Those are serious concerns, while the study succeeds in elegantly connecting the pieces of previous findings into a single story.

The main novelty of this study is that p53 regulates inflammatory cascades via LIF induction and in addition, p53 expression in ISC protects ISC from T cell attack, thus plays a critical role in keeping the small intestinal homeostasis post TBI. These findings would complete the overall picture of one of the intestinal inflammatory cascades that have been proposed by the authors and others. A more detailed mechanistic analysis would increase the suitability of this manuscript.

1. In Fig. 2, viability of p53^{-/-} organoid is decreased even in the absence of DC and differentiated IECs, which seems to suggest the loss of p53 in ISCs alters the sensitivity of ISCs to T cells. If so, what is the underlying mechanisms? Is it related to the lack of p53-mediated apoptosis and induction of mitotic catastrophe? Or do the authors want to imply that priming of ISC by prior stimuli (such as decreased LIF in p53^{-/-}, as discussed in the later part of the manuscript) before crypt isolation is involved in the vulnerability of p53^{-/-} organoids? In either case, this part is a bit confusing and needs to be elaborated.
2. Is MHC-II induced by TBI in crypt cells as well as villi? Also, is there a difference in MHC-II expression between p53^{-/-} and p53^{+/+} ISCs? If ISCs also express MHC-II and its expression is retained within the organoids, this should be mentioned somewhere, as it may also affect the results in Fig. 2.
3. Please specify how organoid viability was quantified. In addition, the more objective index of viability, e.g., LDH levels in the culture supernatant in Fig.2 and Fig 6g should be provided.
4. Activation of CD8⁺ T cells are generally induced by MHC- I . However, the activated CD8⁺ T cells were also induced in irradiated p53^{-/-} (Fig 1g, Fig3c, Fig 4h) and can affect the result of Fig1i. The authors should discuss why activated CD8⁺ T cells were induced and their impact on ISC damage.
5. Mechanism underlying the impairment of the ISC by T cells activated by MHC-II-expressing IEC is unclear in the manuscript. Presenting the number of activated T cells alone appear insufficient, as the number of CD25⁺ or CD69⁺ CD4⁺T cells in p53^{-/-} post TBI is comparable to that of the animals before TBI without inflammation. The markedly increased TNF α levels in p53^{-/-} SI (Fig1h&Fig3d) may reflect the enhanced T cell activation in p53^{-/-}, but may also reflect higher number of macrophage in p53^{-/-}. Therefore, the authors are asked to show secretion of TNF α or other inflammatory cytokines from T cells that can directly or indirectly damage ISCs from T cells in p53^{-/-} mice in comparison to p53^{+/+}.
6. Related to the previous comment, does the rate of TNF α secretion from myeloid cells change between p53^{-/-} and p53^{+/+}?
7. Related to comment 1, Fig. 6g may indicate the effect of p53 on ISCs independent of inflammatory response via the IL12p40/MHC-II axis, and this protective effect is enhanced by LIF. Again, this point is not clearly written in the current manuscript and the authors need to explain more detail what is demonstrated in this T cell-organoid coculture experiment.
8. Fig1b. Please specify which group of genes are up-/down-regulated. (e.g., Down-regulated genes in KO)

Reviewer #1:

The authors of the submitted manuscript investigate the intestinal axis P53/MHCII/IL-12 during the response to radiation injury. The authors demonstrated that WT mice have better survival rates after total-body irradiation than p53^{-/-} mice. They showed that WT mice response is accompanied by earlier and shorter apoptosis in the crypts and efficient regeneration of the crypt compartment. They showed that p53^{-/-} mice have significantly increased and prolonged inflammatory responses compared to WT mice. Furthermore, they demonstrated that immune cells lacking p53 increase inflammation and p53^{-/-} mice have DC cells expressing IL-12-p40 upon TBI. Utilizing multiple in vivo and in vitro models, they showed that p53 regulates IL-12-p40 expression upon radiation injury. The submitted manuscript is exciting, and I have several comments.

Major.

1. There have been studies showing p53/IL-12/p40 in gut regeneration that should be discussed and addressed: PMID 27141366.

Response: Thank the reviewer for pointing out this relevant study (PMID 27141366). As indicated in this study, in the context of melanoma, p53 enhances the production of IL12 in APCs to promote the anti-cancer cytotoxicity. The dual effects of p53 on IL12 production in APCs under different conditions reflect that p53 exerts its function in a highly context-dependent manner which leads to increased anti-cancer immune response towards cancer cells and reduced inflammation in the GI tract to limit radiation-induced GI toxicity, respectively. As suggested by the reviewer, we have added the above discussion on this study in the Discussion part (Page 20, line 416-421).

2. The authors should improve the quality of immunofluorescence and immunohistochemistry images. In addition, the authors should include insets with a higher magnification.

Response: Thank the reviewer for this very valuable suggestion. As suggested by the reviewer, we have replaced the immunofluorescence (IF) and immunohistochemistry (IHC) images with

bigger sizes and higher qualities, and included higher magnification insets for IHC images. As a result, we split both **Figs. 1** and **4** in the original submission into two figures, respectively, to provide enough space to present the IF and IHC images with bigger sizes.

3. In several places, the authors refer to the results obtained from the p53^{-/-} mouse model as proof that intestinal stem cells are only responsible for the regenerative capacity of the gut. Now, there have been many papers showing that multiple cells within the crypts participate in the regeneration and can originate from very different cell populations. I recommend the authors address it or rewrite appropriate sections that, at this stage, are a bit far-reaching.

Response: Thank the reviewer for pointing out the relevance of other cell types contributing to intestinal regeneration upon injury. As suggested by the reviewer, we have added the following statement in the Discussion section: “It is worth noting that the intestinal epithelium shows remarkable flexibility upon damage. It appears that a variety of additional cell types in addition to Lgr5⁺ ISCs can acquire stem cell characteristics after injury and are involved in damage-induced epithelial regeneration. For example, quiescent stem cells, which reside in the +4-+6 cell position from the crypt base, are resistant to stress and can repopulate the crypt upon injury, functioning as reserve ISCs ¹. In addition, several lineage-committed progenitor cells, including ATOH1⁺ or DLL-1⁺ secretory progenitors or ALPI⁺ enterocyte progenitors, Paneth cells, Enteroendocrine cells and Tuft cells have all been shown to be able to acquire stemness property and contribute to injury-induced regeneration ^{2, 3, 4, 5, 6, 7}. For many of the cell types discussed above, the reversion to an Lgr5⁺ ISC state is required for their contribution to the intestinal epithelial regeneration ^{8,9}. Future studies are needed to characterize the effect of p53 on the regenerative activity of these different cell types located within and outside of the crypt upon injury” (Page 20-21, line 431-442).

4. I am unsure whether the authors showed that MHC II on only intestinal epithelial cells participates in signal transduction.

Response: Thank the reviewer for raising this important question. To investigate whether the regulation of MHC-II by p53 exists in other cell types in addition to intestinal epithelial cells (IECs), we have measured the MHC-II expression on macrophages and dendritic cells (DCs), two major types of antigen presenting cells (APCs), in the LP from p53^{+/+} and p53^{-/-} mice post TBI. We observed no significant effect of p53 or TBI on the expression of MHC-II on both macrophages and DCs in the LP (**Fig. S7**). It is worth noting that while macrophages and DCs are the main types of APCs in the immune system, their number in the small intestine and the LP is very limited, especially when it is compared with the number of IECs. Taken together, results from this study strongly suggest that the down-regulation of MHC-II on IECs by p53 plays a major role in ameliorating the inflammation in the GI tract to protect against radiation-induced GI syndrome (Page 8, line 157-160).

5. In addition, the results presented in Figure 7A still need to be completed. As irradiation and p53 are significant players in the suggested model, the authors should show whether rLIF plays a role in irradiation.

Response: Thank the reviewer for this great suggestion. While results in **Fig. 7A** (**Fig. 9A** in the revised manuscript) show that transient rLIF administration has no obvious effect on intestinal tumor growth in *Apc*^{Min/+} mice, it is unclear whether rLIF retains its protective role against radiation-induced GI syndrome in *Apc*^{Min/+} mice. As suggested by the reviewer, we carried out additional experiments to examine the potential role of rLIF in response to irradiation in *Apc*^{Min/+} mice. *Apc*^{Min/+} mice at the age of 120 days (when tumors are fully developed) were subjected to 12 Gy TBI, and rLIF or PBS were injected 3 days before till 3 days post TBI (the same regimen that protects against radiation-induced GI damage in WT mice). rLIF treatment significantly promoted the lifespan of *Apc*^{Min/+} mice post TBI, highlighting the potential application of rLIF in preventing radiation-induced GI syndrome in cancer patients (**Fig. 9b**, page 17, line 359-362). Thus, these experiments suggested by the reviewer significantly strengthen the translational potential of our findings.

Reviewer #3:

In this manuscript, Wang and coworkers study the mechanisms by which ISC is protected from radiotherapy: p53 protects ISC from radiotherapy by inhibiting IL12p40/MHC-II pathway via induction of LIF. While the study is generally well performed and presented, inflammatory cascade of LIF/IL12p40/MHC-II/T cell axis in another GI syndrome model (Wang J. et al., Blood 2022), the protective outcome of LIF on ISC against radiation (Wang H. et al., Cell Death & Disease

2020), and transcriptional induction of LIF by p53 (Hu W. et al, Nature 2007), have been already described by the authors themselves in their previous studies. Those are serious concerns, while the study succeeds in elegantly connecting the pieces of previous findings into a single story.

The main novelty of this study is that p53 regulates inflammatory cascades via LIF induction and in addition, p53 expression in ISC protects ISC from T cell attack, thus plays a critical role in keeping the small intestinal homeostasis post TBI. These findings would complete the overall picture of one of the intestinal inflammatory cascades that have been proposed by the authors and others. A more detailed mechanistic analysis would increase the suitability of this manuscript.

1. In Fig. 2, viability of p53^{-/-} organoid is decreased even in the absence of DC and differentiated IECs, which seems to suggest the loss of p53 in ISCs alters the sensitivity of ISCs to T cells. If so, what is the underlying mechanisms? Is it related to the lack of p53-mediated apoptosis and induction of mitotic catastrophe? Or do the authors want to imply that priming of ISC by prior stimuli (such as decreased LIF in p53^{-/-}, as discussed in the later part of the manuscript) before crypt isolation is involved in the vulnerability of p53^{-/-} organoids? In either case, this part is a bit confusing and needs to be elaborated.

Response: Thank the reviewer for raising this important question and providing a very valuable suggestion. As suggested by the reviewer, we have compared the expression levels of LIF between p53^{+/+} and p53^{-/-} intestinal organoids using real-time PCR assays. Compared with p53^{+/+} intestinal organoids, p53^{-/-} intestinal organoids showed significantly reduced LIF expression levels (**Fig. S13**). Importantly, as shown in **Fig. 8g**, rLIF treatment significantly reduced T-cell triggered cell death in both p53^{+/+} and p53^{-/-} intestinal organoids, and displayed a much more pronounced effect in p53^{-/-} organoids, strongly suggesting that the regulation of LIF expression by p53 mediates the sensitivity of intestinal organoids towards T cell-induced damage. In addition, we noticed that rLIF did not provide complete protection T cell-induced damage in p53^{-/-} intestinal organoids, suggesting that other mechanisms may also make a relatively modest contribution in this organoid co-culture model. Data and the explanation of these observations have now been added to the manuscript (Page 16, line 323-326 & 330-332, & page 20, line 421-427).

2. Is MHC-II induced by TBI in crypt cells as well as villi? Also, is there a difference in MHC-II expression between p53^{-/-} and p53^{+/+} ISCs? If ISCs also express MHC-II and its expression is retained within the organoids, this should be mentioned somewhere, as it may also affect the results in Fig. 2.

Response: Thank the reviewer for this very good suggestion of comparing MHC-II levels between p53^{+/+} and p53^{-/-} cells in both crypts and villi. MHC-II levels in the SI from p53^{+/+} and p53^{-/-} mice

with or without TBI were examined by employing IF staining assays using a MHC-II antibody. Consistent with the results obtained by flow cytometric assays (**Fig. 1f** in the original submission, **Fig. 2b** in the revised manuscript), the expression of MHC-II on IECs was increased in the SI in both $p53^{+/+}$ and $p53^{-/-}$ mice post TBI, and the induction of MHC-II on IECs was much more pronounced in the SI in $p53^{-/-}$ mice post TBI (**Fig. S6**). Further, the increase of MHC-II induced by TBI was observed in both villus and crypt cells from $p53^{+/+}$ and $p53^{-/-}$ mice, and the induction of MHC-II was much more pronounced in both villus and crypt cells from $p53^{-/-}$ mice (**Fig. S6**). These data have now been added to the manuscript (Page 8, line 150-151).

The experiment performed in **Fig. 2** in the original submission (**Fig. 3** in the revised manuscript) compared the sensitivity of p53^{+/+} and p53^{-/-} intestinal organoids towards activated T cell-induced damage. The T cells employed in this set of experiments were already activated *in vitro* by PMA and ionomycin before the co-culture, and were actively producing cytokines such as TNF α and IFN γ that mediate the intestinal organoid damage in the co-culture model. Therefore, the levels of MHC-II on ISCs and other intestinal epithelial cells in the organoids are not expected to affect the activity of T cell population that were pre-activated. Both p53^{+/+} and p53^{-/-} organoids were treated with aliquots of equal numbers of the same T cells with the same activity. Thus, results in **Fig. 2** in the original submission (**Fig. 3** in the revised manuscript) showed that p53^{-/-} organoids were much more susceptible to activated T cell-induced damage, which was independent of the regulation of MHC-II on IECs by p53. We have added these description and discussions to the manuscript (Page 9-10, line 184-187 & page 20, line 421-427).

3. Please specify how organoid viability was quantified. In addition, the more objective index of viability, e.g., LDH levels in the culture supernatant in Fig.2 and Fig 6g should be provided.

Response: The organoid viability was quantified by manually counting the percentage of viable organoids in all organoids in each well as described previously^{10, 11}. Opaque organoids with condensed structures or those that have lost adherence were counted as dead.

Thank the reviewer for this very valuable suggestion of measuring LDH levels in the culture supernatant as a more objective index of organoid viability. As suggested, we repeated experiments shown in **Fig. 2** and **Fig. 6g** in the original submission (**Fig. 3** and **Fig. 8g** in the revised manuscript, respectively) and quantified the organoid viability using the CyQUANT™ LDH Cytotoxicity Assay Kit. We have obtained the results LDH levels that were consistent with the microscopic viability data presented previously. We have updated organoid viability quantification data in revised **Fig. 3** and **Fig. 8g**.

4. Activation of CD8+ T cells are generally induced by MHC-I. However, the activated CD8+ T cells were also induced in irradiated p53-/- (Fig 1g, Fig3c, Fig 4h) and can affect the result of Fig1i. The authors should discuss why activated CD8+ T cells were induced and their impact on ISC damage.

Response: Thank the reviewer for raising this very important question. As pointed out by the reviewer, activation of CD8⁺ T cells are generally induced by MHC-I-mediated antigen presentation in immune responses. It is also well-established that activated CD4⁺ helper T cells play a critical role in the full activation of CD8⁺ T cells in response to many viral infections. Specifically in the model systems of the present study, CD4⁺ T cells, which are activated by MHC-II on intestinal epithelial cells (IECs), can further activate CD8⁺ T cells through direct interaction and cytokine secretion^{12, 13}, which explains the observations in **Fig. 1g, 3c, and 4h** (**Fig. 2c, 4c, and 6c** in the revised manuscript) showing that the regulation of MHC-II levels on IECs by p53 affects the activation of both CD4⁺ and CD8⁺ T cells. Both activated CD4⁺ and CD8⁺ T cells can secrete TNF α (**Fig. S8**), contributing to the intestinal inflammation and ISC damage. The higher number of activated CD4⁺ and CD8⁺ T cells in the LP of p53^{-/-} mice leads to more severe radiation-induced GI toxicity in p53^{-/-} mice. We have added these description and discussions to the Discussion (Page 19, line 405-409).

5. Mechanism underlying the impairment of the ISC by T cells activated by MHC-II-expressing IEC is unclear in the manuscript. Presenting the number of activated T cells alone appear insufficient, as the number of CD25+ or CD69+ CD4+T cells in p53^{-/-} post TBI is comparable to that of the animals before TBI without inflammation. The markedly increased TNF α levels in p53^{-/-} SI (Fig1h&Fig3d) may reflect the enhanced T cell activation in p53^{-/-}, but may also reflect higher number of macrophage in p53^{-/-}. Therefore, the authors are asked to show secretion of TNF α or other inflammatory cytokines from T cells that can directly or indirectly damage ISCs from T cells in p53^{-/-} mice in comparison to p53^{+/+}.

Response: Thank the reviewer for raising this very valuable suggestion. We measured the secretion of TNF α from T cells in the LP from p53^{+/+} and p53^{-/-} mice post TBI using flow cytometric assays. TBI significantly induced TNF α secretion from CD4⁺ and CD8⁺ T cells in the LP from p53^{-/-} mice but not p53^{+/+} mice (**Fig. S8**), supporting that the higher MHC-II expression on IECs in p53^{-/-} mice leads to enhanced T cell activation in the LP post TBI (Page 9, line 165-168).

6. Related to the previous comment, does the rate of TNF α secretion from myeloid cells change between p53^{-/-} and p53^{+/+}?

Response: Thank the reviewer for raising this very important question. We have also measured the secretion of TNF α from myeloid cells (macrophages and DCs) in the LP from p53^{+/+} and p53^{-/-} mice post TBI using flow cytometric assays. Macrophages and DCs in the LP showed no obvious difference in TNF α secretion between p53^{+/+} and p53^{-/-} mice post TBI (**Fig. S8**, page 9, line 165-168).

7. Related to comment 1, Fig. 6g may indicate the effect of p53 on ISCs independent of inflammatory response via the IL12p40/MHC-II axis, and this protective effect is enhanced by LIF.

Again, this point is not clearly written in the current manuscript and the authors need to explain more detail what is demonstrated in this T cell-organoid coculture experiment.

Response: We totally agree with the interpretation of the reviewer on results in **Fig. 6g** in the original submission (**Fig. 8g** in the revised manuscript). Results in **Fig. 2** and **Fig. 6g** in the original submission (**Fig. 3** and **Fig. 8g** in the revised manuscript, respectively) and **Fig. S13** obtained during revision indicate that in the T cell-organoid co-culture system, p53 protects ISC from activated T cell-induced damage which is largely mediated by its transcriptional regulation of LIF and is independent of the IL12p40/MHC-II axis, as we discussed in the response to comment #1 and comment #2 above. As suggested by the reviewer, we have added the following statement in the Discussion part: “It’s worth noting that p53^{-/-} intestinal organoids showed significantly reduced levels of *LIF* expression, which is associated with increased vulnerability to T cell-induced damage, and rLIF treatment displayed a much more pronounced effect on reducing T cell-triggered cell death in p53^{-/-} organoids than p53^{+/+} organoids when co-cultured with equal numbers of activated T cells. These results suggest that in addition to down-regulating IL12 production on DCs and the resultant MHC-II expression on IECs, p53 also directly protects ISCs from activated T cell-induced damage through its transcriptional regulation of LIF (**Fig. 9c**)” (Page 20, line 421-427).

8. Fig1b. Please specify which group of genes are up-/down-regulated. (e.g., Down-regulated genes in KO).

Response: Thank the reviewer for pointing out this unclear labeling. We have specified the labeling as “Down-regulated genes in p53^{-/-} SI” and “Up-regulated genes in p53^{-/-} SI” in **Fig. 1b**.

Again, we want to thank the reviewers for their great efforts and very insightful and constructive comments to improve our manuscript. We hope that with these changes and added experiments, our manuscript would be acceptable for publication. Thank you very much!

Sincerely yours,

Wenwei Hu, MD, PhD
Professor
Rutgers Cancer Institute of New Jersey
Rutgers-State University of New Jersey
New Brunswick, NJ 08903, USA
Email: wh221@cinj.rutgers.edu

References

1. Bankaitis ED, Ha A, Kuo CJ, Magness ST. Reserve Stem Cells in Intestinal Homeostasis and Injury. *Gastroenterology* **155**, 1348-1361 (2018).
2. Hendel SK, Kellermann L, Hausmann A, Bindslev N, Jensen KB, Nielsen OH. Tuft Cells and Their Role in Intestinal Diseases. *Front Immunol* **13**, 822867 (2022).
3. Tomic G, *et al.* Phospho-regulation of ATOH1 Is Required for Plasticity of Secretory Progenitors and Tissue Regeneration. *Cell Stem Cell* **23**, 436-443 e437 (2018).
4. van Es JH, *et al.* Dll1+ secretory progenitor cells revert to stem cells upon crypt damage. *Nat Cell Biol* **14**, 1099-1104 (2012).
5. Santos AJM, Lo YH, Mah AT, Kuo CJ. The Intestinal Stem Cell Niche: Homeostasis and Adaptations. *Trends Cell Biol* **28**, 1062-1078 (2018).
6. Atanga R, Singh V, In JG. Intestinal Enteroendocrine Cells: Present and Future Druggable Targets. *Int J Mol Sci* **24**, (2023).
7. Yu S, *et al.* Paneth Cell Multipotency Induced by Notch Activation following Injury. *Cell Stem Cell* **23**, 46-59 e45 (2018).
8. Meyer AR, Brown ME, McGrath PS, Dempsey PJ. Injury-Induced Cellular Plasticity Drives Intestinal Regeneration. *Cell Mol Gastroenterol Hepatol* **13**, 843-856 (2022).
9. Metcalfe C, Kljavin NM, Ybarra R, de Sauvage FJ. Lgr5+ stem cells are indispensable for radiation-induced intestinal regeneration. *Cell Stem Cell* **14**, 149-159 (2014).
10. Grabinger T, *et al.* Ex vivo culture of intestinal crypt organoids as a model system for assessing cell death induction in intestinal epithelial cells and enteropathy. *Cell Death Dis* **5**, e1228 (2014).
11. Matsuzawa-Ishimoto Y, *et al.* An intestinal organoid-based platform that recreates susceptibility to T-cell-mediated tissue injury. *Blood* **135**, 2388-2401 (2020).
12. Castellino F, Germain RN. Cooperation between CD4+ and CD8+ T cells: when, where, and how. *Annu Rev Immunol* **24**, 519-540 (2006).
13. Laidlaw BJ, Craft JE, Kaech SM. The multifaceted role of CD4(+) T cells in CD8(+) T cell memory. *Nat Rev Immunol* **16**, 102-111 (2016).

REVIEWERS' COMMENTS

Reviewer #1 (Remarks to the Author):

The authors successfully addressed my comments.

Reviewer #2 (Remarks to the Author):

The authors have addressed most of the comments and I am happy with the responses made. I have no more comments to make with the current draft.

Reviewer #3 (Remarks to the Author):

The authors well responded to the reviewer's comments, and the manuscript is now improved.

Reviewer #1:

The authors successfully addressed my comments.

Reviewer #2:

The authors have addressed most of the comments and I am happy with the responses made. I have no more comments to make with the current draft.

Reviewer #3:

The authors well responded to the reviewer's comments, and the manuscript is now improved.

Response: Thank all three reviewers for very positive comments.

Again, we want to thank reviewers for their great efforts and very insightful and constructive comments to improve our manuscript. Thank you very much!

Sincerely yours,

Wenwei Hu, MD, PhD
Professor
Rutgers Cancer Institute of New Jersey
Rutgers-State University of New Jersey
New Brunswick, NJ 08903, USA
Email: wh221@cinj.rutgers.edu